# Revisiting Link Prediction: a data perspective

**Haitao Mao**[1],***Juanhui Li**[1], **Harry Shomer**[1], **Bingheng Li**[2],
**Wenqi Fan**[3], **Yao Ma**[4], **Tong Zhao**[5], **Neil Shah**[5] and **Jiliang Tang**[1]
[1]Michigan State University     [2] The Chinese University of Hong Kong, Shenzhen
[3] Hong Kong Polytechnic University     [4] Rensselaer Polytechnic Institute     [5]Snap Inc.
{haitaoma,lijuanh1,shomerha,tangjili}@msu.edu, libingheng@cuhk.edu.cn
wenqi.fan@polyu.edu.hk,may13@rpi.edu,{tzhao, nshah}@snap.com

## Abstract

Link prediction, a fundamental task on graphs, has proven indispensable in various applications, e.g., friend recommendation, protein analysis, and drug interaction prediction. However, since datasets span a multitude of domains, they could have distinct underlying mechanisms of link formation. Evidence in existing literature underscores the absence of a universally best algorithm suitable for all datasets. In this paper, we endeavor to explore principles of link prediction across diverse datasets from a data-centric perspective. We recognize three fundamental factors critical to link prediction: local structural proximity, global structural proximity, and feature proximity. We then unearth relationships among those factors where (i) global structural proximity only shows effectiveness when local structural proximity is deficient. (ii) The incompatibility can be found between feature and structural proximity. Such incompatibility leads to GNNs for Link Prediction (GNN4LP) consistently underperforming on edges where the feature proximity factor dominates. Inspired by these new insights from a data perspective, we offer practical instruction for GNN4LP model design and guidelines for selecting appropriate benchmark datasets for more comprehensive evaluations.

## 1 Introduction

Graphs are essential data structures that use links to describe relationships between objects. Link prediction, which aims to find missing links within a graph, is a fundamental task in the graph domain. Link prediction methods aim to estimate proximity between node pairs, often under the assumption that similar nodes are inclined to establish connections. Originally, heuristic methods (Zhou et al., 2009; Katz, 1953) were proposed to predict link existence by employing handcrafted proximity features to extract important ***data factors***, e.g., local structural proximity and feature proximity. For example, Common Neighbors(CN) algorithm (Zhou et al., 2009) assumes that node pairs with more overlapping between one-hop neighbor nodes are more likely to be connected. To mitigate the necessity for handcrafted features, Deep Neural Networks are utilized to automatically extract high-quality proximity features. In particular, Graph Neural Networks (GNNs) (Kipf & Welling, 2017; 2016; Hamilton et al., 2017) become increasingly popular owing to their excellence in modeling graph data. Nonetheless, vanilla GNNs fall short in capturing pairwise structural information (Zhang et al., 2021; Liang et al., 2022), e.g., neighborhood-overlapping features, achieving modest performance in link prediction. To address these shortcomings, Graph Neural Networks for Link Prediction(GNN4LP)(Zhang & Chen, 2018; Wang et al., 2022; Chamberlain et al., 2023) are proposed to incorporate different inductive biases revolving on pairwise structural information.

New designs on GNN4LP models strive to improve vanilla GNN to capture diverse pairwise data patterns, e.g., local structural patterns (Yun et al., 2021; Wang et al., 2023), the number of paths (Zhu et al., 2021b), and structural position (Zhang & Chen, 2018). These models have found wide applicability across a myriad of real-world graph problems from multiple domains, e.g., paper recommendation, drug interaction prediction, and protein analysis (Kovács et al., 2019; Hu et al., 2020). A recent benchmark (Li et al., 2023) evaluates the performance of GNN4LP models on datasets from diverse domains, and finds performance disparity as there is no universally best-performing

---

*Work was partially done while the author was a research assistant at The Hong Kong Polytechnic University.

GNN4LP model, observing that even vanilla GCN can achieve best performance on certain datasets. (AbuOda et al., 2020; Chakrabarti, 2022) reveal similar phenomena across heuristic algorithms. We conjecture the main reasons for such phenomena are that **(i)** From a model perspective, different models often have preferred data patterns due to their distinct capabilities and inductive biases. **(ii)** From a data perspective, graphs from different domains could originate from distinct underlying mechanisms of link formation. Figure 1 illustrates this disparity in the number of CNs on multiple benchmark datasets[1]. Notably, edges in the OGBL-PPA and OGBL-DDI datasets tend to have many CNs. Considering both model and data perspectives, performance disparity becomes evident where certain models perform well when their preferred data patterns align with particular data mechanisms on particular datasets, but others do not. This suggests that both model and data perspectives are significant to the success of link prediction. While mainstream research focuses on designing better models (Zhang & Chen, 2018; Zhang et al., 2021), we opt to investigate a data-centric perspective on the development of link prediction. Such a perspective can provide essential guidance on model design and benchmark dataset selection for comprehensive evaluation.

To analyze link prediction from a data-centric perspective, we must first understand the underlying data factors across different datasets. To achieve these goals, our study proceeds as follows: **(i)** Drawing inspiration from well-established literature (Huang et al., 2015; McPherson et al., 2001) in network analysis, we pinpoint three key data factors for link prediction: local structural proximity, global structural proximity, and feature proximity. Comprehensive empirical analyses confirm the importance of these three factors. **(ii)** In line with empirical analysis, we present a latent space model for link prediction, providing theoretical guarantee on the effectiveness of the empirically identified data factors. **(iii)** We conduct an in-depth analysis of relationships among data factors on the latent space model. Our analysis reveals the presence of incompatibility between feature proximity and local structural proximity.

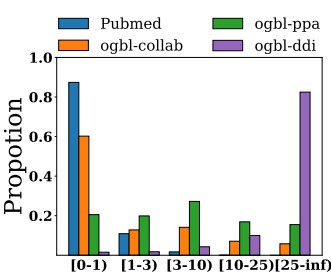

Figure 1: Distribution disparity of Common Neighbors across datasets.

This suggests that the occurrence of both high feature similarity and high local structural similarity within a single edge rarely happens. Such incompatibility sheds light on an overlooked vulnerability in GNN4LP models: they typically fall short in predicting links that primarily arise from feature proximity. **(iv)** Building upon the systematic understandings, we provide guidance for model design and benchmark dataset selection, with opportunities for link prediction.

## 2 RELATED WORK

Link prediction aims to complete missing links in a graph, with applications ranging from knowledge graph completion (Nickel et al., 2015) to e-commerce recommendations (Huang et al., 2005). While heuristic algorithms were once predominant, Graph Neural Networks for Link Prediction (GNN4LP) with deep learning techniques have shown superior performance in recent years.

**Heuristic algorithms**, grounded in the principle that similar nodes are more likely to connect, encompass local structural heuristics like Common Neighbor and Adamic Adar (Adamic & Adar, 2003), global structural heuristics like Katz and SimRank (Jeh & Widom, 2002; Katz, 1953), and feature proximity heuristics(Nickel et al., 2014; Zhao et al., 2017) integrating additional node features.

**GNN4LP** is built on basic GNNs (Kipf & Welling, 2016; 2017) which learn single node structural representation by aggregating neighborhood and transforming features recursively, equipped with pairwise decoders. GNN4LP models augment vanilla GNNs by incorporating more complicated pairwise structural information inspired by heuristic methods. For instance, NCNC (Wang et al., 2023) and NBFNet (Zhu et al., 2021b) generalize CN and Katz heuristics with neural functions to incorporate those pairwise information, thereby achieving efficiency and promising performance. A more detailed discussion on heuristics, GNNs, and principles in network analysis is in Appendix A.

---

[1]More evidence on other data properties can be found in Appendix E.

## 3 MAIN ANALYSIS

In this section, we conduct analyses to uncover the key data factors for link prediction and the underlying relationships among those data factors. Since underlying data factors contributing to link formation are difficult to directly examine from datasets, we employ heuristic algorithms as a lens to reflect their relevance. Heuristic algorithms calculate similarity scores derived from different data factors to examine the probability of whether two nodes should be connected. They are well-suited for this analysis as they are simple and interpretable, rooted in principles from network analysis (Murase et al., 2019; Khanam et al., 2020). Leveraging proper-selected heuristic algorithms and well-established literature in network analysis, we endeavor to elucidate the underlying data factors for link prediction.

**Organization.** Revolving on the data perspective for link prediction, the following subsections are organized as follows. Section 3.1 focuses on identifying and empirically validating the key data factors for link prediction using corresponding heuristics. In line with the empirical significance of those factors, Section 3.2 introduces a theoretical model for link prediction, associating data factors with node distances within a latent space. Links are more likely to be established between nodes with a small latent distance. Section 3.3 unveils the relationship among data factors building upon the theoretical model. We then clearly identify an incompatibility between local structural proximity and feature proximity factors. Specifically, incompatibility indicates it is unlikely that the occurrence of both large feature proximity and large local structural proximity within a single edge. Section 3.4 highlights an overlooked limitation of GNN4LP models stemming from this incompatibility.

**Preliminaries & Experimental Setup.** $\mathcal{G} = (\mathcal{V}, \mathcal{E})$ is an undirected graph where $\mathcal{V}$ and $\mathcal{E}$ are the set of $N$ nodes and $|\mathcal{E}|$ edges, respectively. Nodes can be associated with features $\mathbf{X} \in \mathbb{R}^{n \times d}$, where $d$ is the feature dimension. We conduct analysis on CORA, CITESEER, PUBMED, OGBL-COLLAB, OGBL-PPA, and OGBL-DDI datasets (Hu et al., 2020; McCallum et al., 2000) with the same model setting as recent benchmark (Li et al., 2023). Experimental and dataset details are in Appendix K and J, respectively.

### 3.1 UNDERLYING DATA FACTORS ON LINK PREDICTION

Motivated by well-established understandings in network analysis (Daud et al., 2020; Wang & Le, 2020; Kumar et al., 2020) and heuristic designs (Adamic & Adar, 2003; Katz, 1953), we conjecture that there are three key data factors for link prediction.

**(1)** `Local structural proximity(LSP)` (Newman, 2001) corresponds to the similarity of immediate neighborhoods between two nodes. The rationale behind LSP is rooted in the principle of triadic closure (Huang et al., 2015), which posits that two nodes with more common neighbors have a higher probability of being connected. Heuristic algorithms derived from the LSP perspective include CN, RA, and AA (Adamic & Adar, 2003), which quantify overlap between neighborhood node sets. We mainly focus on common neighbors (CN) in the following discussion. The CN score for nodes $i$ and $j$ is calculated as $|\Gamma(i) \cap \Gamma(j)|$, where $\Gamma(\cdot)$ denotes the neighborhood set. More analysis on other related heuristics revolving around LSP, e.g., RA, AA, can be found in Appendix C.5.

**(2)** `Global structural proximity(GSP)` (Katz, 1953; Jeh & Widom, 2002) goes beyond immediate neighborhoods between two nodes by considering their global connectivity. The rationale behind GSP is that two nodes with more paths between them have a higher probability of being connected. Heuristic algorithms derived from GSP include SimRank, Katz, and PPR (Brin & Page, 2012), to extract the ensemble of paths information. We particularly focus on the Katz heuristic in the following discussion. The Katz score for nodes $i$ and $j$ is calculated as $\sum_{l=1}^{\infty} \lambda^l |\text{paths}^{\langle l \rangle}(i,j)|$, where $\lambda < 1$ is a damping factor, indicating the importance of the higher-order information. $|\text{paths}^{\langle l \rangle}(i,j)|$ counts the number of length-$l$ paths between $i$ and $j$.

**(3)** `Feature proximity(FP)` (Murase et al., 2019) corresponds to the feature similarity between nodes. The rationale behind FP is the principle of feature homophily (Khanam et al., 2020; Evtushenko & Kleinberg, 2021), which posits two nodes with more similar individual characteristics have a higher probability of being connected. There are many heuristic algorithms (Tang et al., 2013; Zhao et al., 2017) derived from the FP perspective. Nonetheless, most of them combine FP in addition to the above structural proximity, leading to difficulties for analyzing FP solely. Hence, we derive a simple heuristic called feature homophily (FH) focusing on only feature proximity solely for ease of

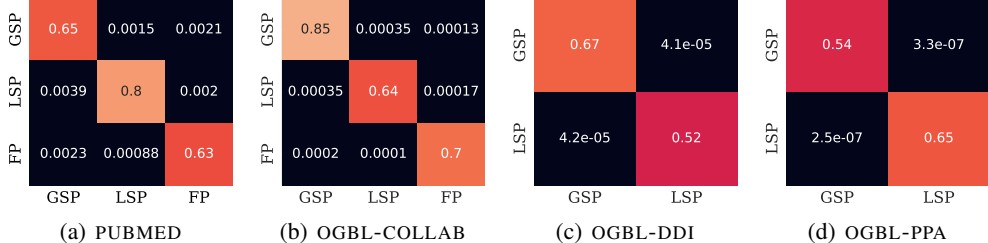

|     | (a) PUBMED | (b) OGBL-COLLAB | (c) OGBL-DDI | (d) OGBL-PPA |

Figure 3: Overlapping ratio between top-ranked edges on different heuristic algorithms. **Diagonals are the comparison between two heuristics within the same factor**, while others compare heuristics from different factors. FP is ignored on OGBL-DDI and OGBL-PPA due to no or weak feature quality. MRR is selected as the metric. More results on hit@10 metric can be found in Appendix D.

analysis. The FH score between nodes $i$ and $j$ is calculated as $\text{dis}(x_i, x_j)$, where $x_i$ corresponds to the node feature, and $\text{dis}(\cdot)$ is a distance function. We particularly focus on FH with cosine distance function in the following discussion. Notably, details on all the heuristics mentioned above can be found in Appendix A and B. To understand the importance of those data factors, we aim to answer the following questions: **(i)** Does each data factor indeed play a key role in link prediction? **(ii)** Does each factor provide unique information instead of overlapping information?

We first concentrate on examining the significance of each afore-mentioned factor for link prediction, based on well-established principles from network analysis. We exhibit the performance of heuristic algorithms in Figure 2. We make the following observations: **(i)** For datasets from the academic domain, CORA, CITESEER, PUBMED, and OGBL-COLLAB, we find that heuristics for different factors can achieve satisfying performance. The Katz corresponding to the GSP factor consistently outperforms other heuristics. Explanations of the phenomenon are further presented in the following Section 3.3. **(ii)** For OGBL-DDI and OGBL-PPA datasets, CN heuristic corresponding to the LSP factor consistently performs best while FH performs poorly. We conjecture that this is due to low feature quality. For instance, node features for OGBL-PPA are a one-hot vector corresponding

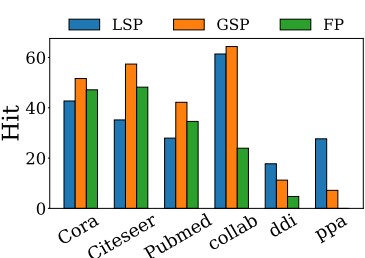

Figure 2: Performance of heuristics corresponding to different factors.

to different categories. **(iii)** No single heuristic algorithm consistently outperforms across all datasets, indicating data disparity; detailed discussions are in Section 4.2. The effectiveness of heuristics is very data-specific which further highlights the importance of investigating link prediction from a data perspective.

We further investigate the relationship between heuristics from the same and different data factors. Details on the heuristic selection are in Appendix B. This includes whether heuristics from the same data factor provide similar information for link prediction and whether those from different data factors could offer unique perspectives. To this end, we examine disparities in predictions among different heuristics. Similar predictions imply that they provide similar information, while divergent predictions indicate that each factor could provide unique information. Predictions for node pairs can be arranged in descending order according to the predicted likelihood of them being connected. We primarily focus on top-ranked node pairs since they are likely to be predicted as links. Thus, they can largely determine the efficacy of the corresponding algorithm. If two algorithms produce similar predictions, high-likelihood edges should have a high overlap. Else, their overlap should be low.

Experimental results are shown in Figure 3. Due to the low feature quality, we exclude OGBL-DDI and OGBL-PPA datasets as we conduct analyses on all three factors. It focuses on the overlapping ratio between the top ranking (25%) node pairs of two different heuristics either from the same data factor or different data factors. We make the following observations: **(i)** Comparing two heuristics from the same factor, i.e., the diagonal cells, we observe that high-likelihood edges for one heuristic are top-ranked in the other. This indicates heuristics from the same data factor capture similar information. **(ii)** Comparing two heuristics derived from different factors, We can observe that the overlapping of top-ranked edges is much lower, especially when comparing GSP and FP, as well as LSP and FP.

Though GSP and LSP factors have a relatively high overlapping in top-ranked edges, the overlapping is still much smaller than that for heuristics from the same factor. These observations suggest that **(i)** selecting one representative heuristic for one data factor could be sufficient as heuristics from the same factors share similar predictions, and **(ii)** different factors are unique as there is little overlap in predictions. More analyses are in Appendix E and F.

## 3.2 THEORETICAL MODEL FOR LINK PREDICTION BASED ON UNDERLYING DATA FACTORS

In this subsection, we rigorously propose a theoretical model for link prediction based on the important data factors empirically analyzed above. We first introduce a latent space model and then theoretically demonstrate that the model reflects the effectiveness of LSP, GSP, and FP factors. All proofs can be found in Appendix C given space constraints.

The latent space model (Hoff et al., 2002) has been widely utilized in many domains, e.g., sociology and statistics. It is typically utilized to describe proximity in the latent space, where nodes with a close in the latent space are likely to share particular characteristics. In our paper, we propose a latent space model for link prediction, describing a graph with $N$ nodes incorporating both feature and structure proximity, where each node is associated with a location in a $D$-dimensional latent space. Intuitions on modeling feature and structure perspectives are shown as follows. **(i)** *Structural perspective* is of primarily significance in link prediction. In line with this, the latent space model connects the link prediction problem with the latent node pairwise distance $d$, where $d$ is strongly correlated with structural proximity. A small $d_{ij}$ indicates two nodes $i$ and $j$ sharing similar structural characteristics, with a high probability of being connected. **(ii)** *Feature perspective* provides complementary information, additionally considering two nodes with high feature proximity but located distantly in the latent space should also be potentially connected. In line with this, we introduce the feature proximity parameter $\beta_{ij}$ in the latent space. A larger $\beta_{ij}$ indicates more likely for node $i$ and $j$ to be connected. Considering feature and structural perspectives together, we develop an undirected graph model inspired by (Sarkar et al., 2011). Detailed formulation is as follows:

$$P(i \sim j|d_{ij}) = \begin{cases} \dfrac{1}{1 + e^{\alpha(d_{ij} - \max\{r_i, r_j\})}} \cdot (1 - \beta_{ij}) & d_{ij} \leq \max\{r_i, r_j\} \\ \beta_{ij} & d_{ij} > \max\{r_i, r_j\} \end{cases} \tag{1}$$

where $P(i \sim j|d_{ij})$ depicts the probability of forming an undirected link between $i$ and $j$ ($i \sim j$), predicated on both the features and structure. The latent distance $d_{ij}$ indicates the structural likelihood of link formation between $i$ and $j$. The feature proximity parameter $\beta_{ij} \in [0,1]$ additionally introduces the influence from the feature perspective. Moreover, the model has two parameters $\alpha$ and $r$. $\alpha > 0$ controls the sharpness of the function. To ease the analysis, we set $\alpha = +\infty$. Discussions on when $\alpha \neq +\infty$ are in Appendix C.6. $r_i$ is a connecting threshold parameter corresponding to node $i$. With $\alpha = +\infty$, $\frac{1}{1 + e^{\alpha(d_{ij} - \max\{r_i, r_j\})}} = 0$ if $d_{ij} > \max\{r_i, r_j\}$, otherwise it equals to 1. Therefore, a large $r_i$ indicates node $i$ is more likely to form edges, leading to a potentially larger degree. Nodes in the graph can be associated with different $r$ values, allowing us to model graphs with various degree distributions. Such flexibility enables our theoretical model to be applicable to more real-world graphs. We identify how the model can reveal different important data factors in link prediction. Therefore, we **(i)** derive heuristic scores revolving around each factor in the latent space and **(ii)** provide a theoretical foundation suggesting that each score can offer a suitable bound for the probability of link formation. Theoretical results underscore the effectiveness of each factor.

**Effectiveness of Local Structural Proximity (LSP).** We first derive the common neighbor (CN) score on the latent space model. Notably, since we focus on the local structural proximity, the effect of the features is ignored. We therefore set the FP parameter $\beta_{ij} = 0$, for ease of analysis. Considering two nodes $i$ and $j$, a common neighbor node $k$ can be described as a node connected to both nodes $i$ and $j$. In the latent space, it should satisfy both $d_{ik} < \max\{r_i, r_k\}$ and $d_{kj} < \max\{r_k, r_j\}$, which lies in the intersection between two balls, $V(\max\{r_i, r_k\})$ and $V(\max\{r_k, r_j\})$. Notably, $V(r) = V(1)r^D$ is the volume of a radius $r$, where $V(1)$ is the volume of a unit radius hypersphere. Therefore, the expected number of common neighbor nodes is proportional to the volume of the intersection between two balls. Detailed calculations are in Appendix C.1. With the volume in the latent space, we then derive how CN provides a meaningful bound on the structural distance $d_{ij}$.

**Proposition 1** (latent space distance bound with CNs). *For any $\delta > 0$, with probability at least $1 - 2\delta$, we have $d_{ij} \leq 2\sqrt{r_{ij}^{max} - \left(\frac{\eta_{ij}/N - \epsilon}{V(1)}\right)^{2/D}}$, where $\eta_{ij}$ is the number of common neighbors between nodes $i$ and $j$, $r_{ij}^{max} = max\{r_i, r_j\}$, and $V(1)$ is the volume of a unit radius hypersphere in $D$ dimensional space. $\epsilon$ is a term independent of $\eta_{ij}$. It vanishes as the number of nodes $N$ grows.*

Proposition 1 indicates that a large number of common neighbors $\eta_{ij}$ results in a smaller latent distance $d_{ij}$, leading to a high probability for an edge connection. We then extend the above analysis on local structure to global structure with more complicated structural patterns.

**Effectiveness of Global Structural Proximity (GSP).** We first derive the number of paths between node $i$ and $j$ on the latent space. Notably, most heuristics on the GSP factor can be viewed as a weighted number of paths. The key idea is to view each common neighbor node as a path with a length $\ell = 2$, serving as the basic element for paths with a length $\ell > 2$. We denote that the nodes $i, j$ are linked through path of length $\ell$, i.e., $i = k_0 \sim k_1 \sim \ldots \sim k_{\ell-1} \sim k_\ell = j$. As we assume each node is only associated with its neighborhood, the probability that the path $P(k_0 \sim k_1 \sim \ldots \sim k_{\ell-1} \sim k_\ell)$ exists can be easily bounded by a decomposition of $P(k_0 \sim k_1 \sim k_2) \cdot P(k_1 \sim k_2 \sim k_3) \cdots P(k_{\ell-2} \sim k_{\ell-1} \sim k_\ell) = \prod_{l=1}^{\ell-1} P(k_{\ell-1}, k_\ell, k_{\ell+1})$. Notably, each element is the common neighbor probability discussed in Proposition 1, equivalent to the path with $\ell = 2$. We then calculate the volume of the number of paths and derive how it bound the latent distance $d_{ij}$.

**Proposition 2** (latent space distance bound with the number of paths). *For any $\delta > 0$, with probability at least $1 - 2\delta$, we have $d_{ij} \leq \sum_{n=0}^{M-2} r_n + 2\sqrt{r_M^{\max} - \left(\frac{\eta_\ell(i,j) - b(N,\delta)}{c(N,\delta,\ell)}\right)^{\frac{2}{D(\ell-1)}}}$, where $\eta_\ell(i,j)$ is the number of paths of length $\ell$ between $i$ and $j$ in $D$ dimensional Euclidean space. $M \in \{1, \cdots, \ell - 1\}$ is the set of intermediate nodes.*

Proposition 2 indicates that a large number of paths $\eta_\ell(i,j)$ results in a smaller latent distance $d_{ij}$, leading to a high probability for an edge connection. It demonstrates the effectiveness of GSP factor.

**Effectiveness of Feature Proximity (FP).** We next focus on the role of FP parameter $\beta_{ij}$. In particular, we extend Proposition 1 which ignored the FP with $\beta_{ij} = 0$, to $\beta_{ij} = [0, 1]$. This allows distant nodes in the latent space to be connected with each other if they share similar features. Specifically, two nodes $i, j$ with latent distance $d_{ij} > r_{max}\{r_i, r_j\}$ are connected to each other with a probability $\beta_{ij}$. Instead of defining that there is no edge connected with $p = 0$ when $d_{ij} > max\{r_i, r_j\}$, nodes are instead connected with a probability of $p = \beta_{ij}$. This provides a connection probability for node pairs with high FP. Revolving on the additional consideration of FP, we show the proposition as follows:

**Proposition 3** (latent space distance bound with feature proximity). *For any $\delta > 0$, with probability at least $1 - 2\delta$, we have $d_{ij} \leq 2\sqrt{r_{ij}^{max} - \left(\frac{\beta_{ij}(1 - A(r_i, r_j, d_{ij})) + A(r_i, r_j, d_{ij})}{V(1)}\right)^{2/D}}$, where $\beta_{ij}$ measures feature proximity between $i$ and $j$, $r_{ij}^{max} = max\{r_i, r_j\}$ and $V(1)$ is the volume of a unit radius hypersphere in $D$ dimensional Euclidian space. $A(r_i, r_j, d_{ij})$ is the volume of intersection of two balls of $V(r_i)$ and $V(r_j)$ in latent space, corresponding to the expectation of common neighbors.*

We can observe that when $A(r_i, r_j, d_{ij})$ is fixed, a larger $\beta_{ij}$ leads to a tighter bound with close distance in the latent space. Proposition 3 indicates that a high FP results in a small latent distance $d_{ij}$, leading to a high probability for an edge connection. Notably, the conclusion could easily extend two Proposition 2 on global structural proximity with details in Appendix C.4. The above theoretical results indicate the significance of the three data factors.

### 3.3 Intrinsic relationship among underlying data factors

In this subsection, we conduct a rigorous analysis elucidating the intrinsic relationship among different factors, upon the theoretical model. Our analyses are two-fold: **(i)** the relationship between structural factors, i.e., LSP and GSP; and **(ii)** the relationship between factors focusing on feature and structure, i.e., FP and LSP, FP and GSP. Proof details are in Appendix C.

**The relationship between local and global structural proximity.** To consider both local and global structural factors, we treat the CN algorithm as the number of paths $\eta_\ell(i,j)$ with length $\ell = 2$. Therefore, analysis between local and global structural factors can be regarded as the influence of

$\eta(i, j)$ on different lengths $\ell$. The key for the proof is to identify the effect of $\ell$ by bounding other terms related with $\ell$ in Proposition 2, i.e., $\eta_\ell(i, j)$ and $c(N, \delta, \ell)$. We also ignore the feature effect to ease the structural analysis.

**Lemma 1** (latent space distance bound with local and global structural proximity). *For any $\delta > 0$, with probability at least $1 - 2\delta$, we have $d_{ij} \leq \sum_{n=0}^{M-2} r_n + 2\sqrt{r_M^{max} - \left(\sqrt{\frac{N \ln(1/\delta)}{2}} - 1\right)^{\frac{2}{D(\ell-1)}}}$, where $\sum_{n=0}^{M-2} r_n$, $r_M^{max}$ serve as independent variables that do not change with $\ell$.*

Given the same number of paths $\eta_\ell$ with different lengths $\ell$, a small $\ell$ provides a much tighter bound with close distance in the latent space. The bound becomes exponentially loose with the increase of $\ell$ as the hop $\ell$ in $\left(\sqrt{\frac{N \ln(1/\delta)}{2}} - 1\right)^{\frac{2}{D(\ell-1)}}$ acts as an exponential coefficient. This indicates that **(i)** When both LSP and GSP are sufficient, LSP can provide a tighter bound, indicating a more important role. **(ii)** When LSP is deficient, e.g., the graph is sparse with not many common neighborhoods, GSP can be more significant. The theoretical understanding can also align with our empirical observations in Section 3.1. Figure 2 illustrates that **(i)** heuristics derived from GSP perform better on sparse graphs with deficient common neighbors shown in Figure 1. **(ii)** The heuristics derived from LSP perform better on the dense graph, i.e., OGBL-DDI and OGBL-PPA with more common neighbors.

**The relationship between structural and feature proximity.** Our analysis then focuses on the interplay between feature and structural proximity. The key for the proof is to recognize how feature proximity could affect the number of common neighbors derived from the LSP factor.

**Lemma 2** (Incompatibility between LSP and FP factors). *For any $\delta > 0$, with probability at least $1 - 2\delta$, we have $\eta_{ij} = \frac{c'}{1 - \beta_{ij}} + N(1+\epsilon)$, where $\eta_{ij}$ and $\beta_{ij}$ are the number of common neighbor nodes and feature proximity between nodes $i$ and $j$. $c' < 0$ is an independent variable that does not change with $\beta_{ij}$ and $\eta_{ij}$. $\eta_{ij}$ is negatively correlated with $\beta_{ij}$.*

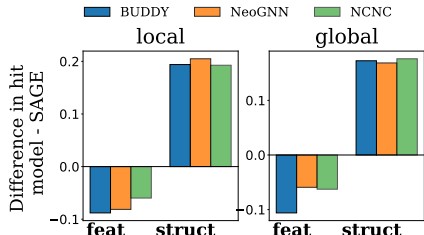

Figure 4: Performance comparison between GNN4LP models and SAGE on the OGBL-COLLAB dataset. Bars represent the performance gap on node pairs dominated by feature and structural proximity, respectively. Figures correspond to compare FP with GSP and LSP, respectively

Lemma 2 demonstrates that node pairs with a large number of common neighbors $\eta_{ij}$ tend to have low feature proximity $\beta_{ij}$ and vice versa. Such findings underscore the incompatibility between LSP and feature proximity, where it is unlikely that both large LSP and FP co-exist in a single node pair. It challenges the conventional wisdom, which posits that LSP tends to connect people, reinforcing existing FP, e.g., connecting people with similar characteristics. However, our findings suggest that LSP could offset the feature proximity. One intuitive explanation of such phenomenon from social network literature (Abebe et al., 2022) is that, in contexts with FP, similar individuals tend to connect. Thus, if nodes with common neighbors (mutual friends) do not have a link connected, their features may be quite different. The new edge forms between those nodes with high LSP actually connect individuals with low FP. A similar relationship is also established between GSP and FP with proof in Appendix C.4.

### 3.4 AN OVERLOOKED VULNERABILITY IN GNN4LP MODELS INSPIRED FROM DATA FACTORS

In this subsection, we delve into how the incompatibility between structural proximity and feature proximity affects the effectiveness of GNN4LP models. These models are inherently designed to learn pairwise structural representation, encompassing both feature and structural proximity. Despite their strong capability, the incompatibility between structural and feature factors leads to potentially conflicting training signals. For example, while structural proximity patterns may imply a likely link between two nodes, feature proximity patterns might suggest the opposite. Therefore, it seems challenging for a single model to benefit both node pairs with feature proximity factor and those with the structural ones. Despite most research primarily emphasizing the capability of GNN4LP models on structural proximity, the influence of incompatibility remains under-explored.

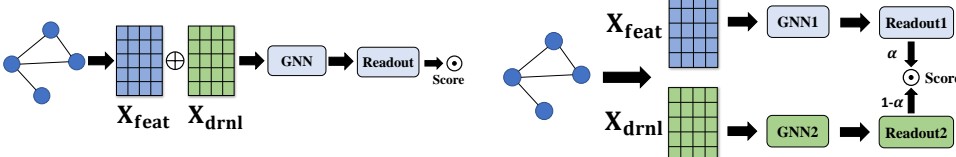

(a) Original coupling SEAL model architecture    (b) Proposed decoupled SEAL model architecture

Figure 5: The original SEAL and the proposed decoupled SEAL architectures. $\mathbf{X}_{\text{feat}}$ and $\mathbf{X}_{\text{drnl}}$ are the original node feature and the structural embedding via Double-Radius Node Labeling.

To validate our statement, we conduct experiments to compare the performance of vanilla GNNs, e.g., SAGE and GCN, with the advanced GNN4LP models including Buddy, NeoGNN, and NCNC. The fundamental difference between GNN4LP models and vanilla GNNs is that vanilla GNNs only learn single-node structural representation with limitations in capturing pairwise structural factors while GNN4LP models go beyond. Such comparison sheds light on examining how the key capacity of GNN4LP, i.e., capturing pairwise structural factor, behaves along the incompatibility. Comparisons are conducted on node pairs dominated by different factors, represented as node pairs $\mathcal{E}_s \setminus \mathcal{E}_f$ and $\mathcal{E}_f \setminus \mathcal{E}_s$ with only structural proximity and only feature proximity accurately predicted, respectively. $\mathcal{E}_s$ and $\mathcal{E}_f$ denote node pairs accurately predicted with structural proximity and feature proximity, respectively. Experimental results are presented in Figure 4, where the x-axis indicates node pairs dominated by different underlying factors. The y-axis indicates the performance differences between GNN4LP models and vanilla GraphSAGE. More results on GCN can be found in Appendix E. A notable trend is that GNN4LP models generally outperform vanilla GNNs on edges governed by LSP and GSP while falling short in those on feature proximity. This underlines the potential vulnerability of GNN4LP models, especially when addressing edges primarily influenced by feature proximity. This underlines the overlooked vulnerability of GNN4LP models on node pairs dominated by the FP factor due to the incompatibility between feature and structural proximity.

## 4 GUIDANCE FOR PRACTITIONERS ON LINK PREDICTION

In this section, we provide guidance for the new model design and how to select benchmark datasets for comprehensive evaluation, based on the above understandings from a data perspective.

### 4.1 GUIDANCE FOR THE MODEL DESIGN

In Section 3, we highlight the incompatibility between structural and feature proximity factors in influencing GNN4LP models. When both structural and feature factors come into play simultaneously, there is a potential for them to provide conflicting supervision to the model. Such understanding suggests that the model design should learn the feature proximity factors and pairwise structural ones independently before integrating their outputs, in order to mitigate such incompatibility. In particular, we apply such a strategy to the SEAL (Zhang & Chen, 2018), a representative GNN4LP model. Different from the vanilla GNNs only utilizing original node features $\mathbf{X}_{feat}$ as feature input, it additionally employs local structural features $\mathbf{X}_{drnl}$ by double-radius node labeling (DRNL) based on their structural roles. $\mathbf{X}_{feat}$ and $\mathbf{X}_{drnl}$ are concatenated and then forwarded to one single GNN, as depicted in Figure 5(a). Therefore, the GNN must wrestle with the incompatibility between FP and structural factors. Guided by the above understanding, we propose the decoupled SEAL, which separates the original node features $\mathbf{X}_{feat}$ and local structural features $\mathbf{X}_{drnl}$ into different GNNs. Each dedicated GNN could learn either feature patterns or pairwise structural patterns separately. The decoupled model architecture is depicted in Figure 5(b). Experimental results comparing the original SEAL and our proposed decoupled SEAL are illustrated in Figure 6(a). Notably, our decoupled

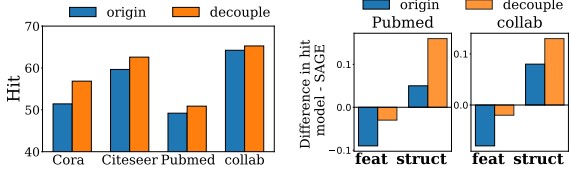

(a) Performance on original and decoupled SEAL.  (b) Comparison between SEAL models and SAGE.

Figure 6: Effectiveness of proposed decoupled SEAL comp.

SEAL consistently outperforms, with gains reaching up to 1.03% on the large OGBL-COLLAB dataset. Furthermore, Figure 6(b) shows comparisons with GraphSAGE, following the same setting with Figure 4. The decoupled SEAL demonstrates a reduced performance drop on node pairs dominated by the FP factor with a larger gain on those by structural factors. Code is available at here.

Table 1: The Hit@10 performance on the newly selected datasets.

|  | CN | Katz | FH | MLP | SAGE | BUDDY |
|---|---|---|---|---|---|---|
| POWER | 12.88 | 29.85 | NA | $5.03 \pm 0.88$ | $6.99 \pm 1.16$ | $19.88 \pm 1.37$ |
| PHOTO | 18.34 | 7.07 | 13.78 | $12.37 \pm 4.13$ | $18.61 \pm 5.97$ | $18.09 \pm 2.52$ |

## 4.2 GUIDANCE FOR BENCHMARK DATASET SELECTION

With the recognized data factors and their relationships, we enumerate all potential combinations among different data factors, illuminating the complete dataset landscape. It allows us to categorize prevalent datasets and pinpoint missing scenarios not covered by those datasets. Consequently, we introduce new datasets addressing those identified gaps and offer guidance for practitioners on more comprehensive benchmark dataset selection. In particular, we recognize datasets into four categories considering two main aspects: **(i)** From the feature perspective, we verify whether FP dominates, indicated with decent performance on FH. **(ii)** From the structural perspective, we verify whether GSP dominates, indicated by whether a GSP heuristic can provide additional improvement over LSP (if not, then LSP dominates). Section 3.3 demonstrates that such scenario happens when LSP is inadequate. Therefore, there are four categories including **category 1**: both LSP and FP factors dominate. **Category 2**: Only LSP factor dominates. **Category 3**: both GSP and FP factors dominate. **Category 4**: Only GSP factor dominates. Evidence in Figure 2 helps to categorize existing benchmarking datasets. The prevalent datasets like CORA, CITESEER, and PUBMED are in category 3 with both GSP and FP factors dominating, while datasets like OGBL-DDI and OGBL-PPA primarily are in category 2, focusing on the LSP factor. We can then clearly identify that two significant dataset categories, 1 and 4, are not covered on existing datasets.

To broaden for a more comprehensive evaluation beyond existing benchmark datasets, we introduce more datasets to cover these categories . This includes the unfeatured POWER dataset in category 4 and the PHOTO dataset in category 1. The categorizations of these datasets are confirmed through experimental results illustrated in Table 1. We observe: **(i)** For the POWER dataset with only GSP matters, the Katz significantly outperforms other algorithms, even the GNN4LP model, BUDDY. **(ii)** Deep models do not show superior performance on both datasets, indicating that success focusing on existing datasets cannot extend to the new ones, suggesting potential room for improvement. We can then provide the following guidance for benchmarking dataset selection for practitioners: **(i)** selecting algorithms that perform best on the datasets belonging to the same category as the proposed one. **(ii)** selecting datasets from their own domain rather than datasets from other domains. To help with that, we collect most of the existing datasets for link prediction covering most domains including biology, transportation, web, academia, and social science, assisting in a more comprehensive evaluation aligning with real-world scenarios. Details on all datasets are in Appendix D and the repository.

## 5 CONCLUSION

In this work, we explore link prediction from a data perspective, elucidating three pivotal factors: LSP, GSP, and FP. Theoretical analyses uncover the underlying incompatibility. Inspired by incompatibility, our paper shows a positive broader impact as we identify the overlooked biased prediction in GNN4LP models and show the potential solution to address this issue. Our understanding provides guidance for the new model design and how to select benchmark datasets for comprehensive evaluation. Such understanding also gains insights for future direction including (1) adding a more careful discussion on the above fairness issue and (2) designing specific GNN4LP models for datasets in different dataset categories mentioned in Sec 4.2. Nonetheless, our paper shows minor limitations as we make the assumption that the feature proximity is an additional noise parameter rather than adaptively combining that information in the same subspace in theoretical analysis. A more comprehensive discussion on Limitation, broader impact, and future works are in Appendix G,H, and I.

## 6 ACKNOWLEDGEMENT

This research is supported by the National Science Foundation (NSF) under grant numbers CNS 2246050, IIS1845081, IIS2212032, IIS2212144, IIS-2406648, IIS-2406647, IOS2107215, DUE 2234015, DRL 2025244 and IOS2035472, the Army Research Office (ARO) under grant number W911NF-21-1-0198, the Home Depot, Cisco Systems Inc, Amazon Faculty Award, Johnson&Johnson, JP Morgan Faculty Award and SNAP.

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

# Appendix

## Table of Contents

## A    RELATED WORK

Link prediction is a fundamental task in graph analysis. It seeks to determine the probability of a connection between two nodes in a graph, leveraging both the existing graph topology and node features. This task plays a critical role in a variety of applications, such as improving knowledge graph completion (Nickel et al., 2015), enhancing metabolic networks (Oyetunde et al., 2017), suggesting potential friends in social media platforms (Adamic & Adar, 2003; Gupta et al., 2013), predicting protein-protein interactions (Kovács et al., 2019), and refining e-commerce recommendations (Huang et al., 2005). Heuristic algorithms initially dominated, serving as the primary methodology, due to their simplicity, interpretability, and scalability. Then, GNNs are introduced which leverages the power of deep learning. It allows for automatic feature extraction with superior performance.

**Heuristic algorithms** are based on the hypothesis that nodes with higher similarity have an increased probability of connection. The proximity score between two nodes represents their potential connection probability. This score is tailored to the characteristics of specific node pairs, viewed from various perspectives. Generally, there are three types of heuristic approaches to measure the proximity and the formal definition of some representative ones are listed in Table 2. (1) **Local structural heuristics.** This category includes four prevalent algorithms: Common Neighbor (CN) (Newman, 2001), Jaccard (Jaccard, 1902), Adamic Adar (AA) (Adamic & Adar, 2003) and Resource Allocation (RA) (Zhou et al., 2009). These methods are fundamentally based on the assumption that nodes sharing more common neighbors are more likely to connect. AA and RA are weighted variants of common Neighbors. Typically, they incorporate node degree information, emphasizing that nodes with a lower degree have a higher influence. (2) **Global structural proximity heuristics.** This category includes algorihms Katz (Katz, 1953), SimRank (Jeh & Widom, 2002), and Personalized PageRank (PPR) (Brin & Page, 2012) in the graph. They consider the global structural patterns with the number of paths between two nodes, where more paths indicate high similarity and are more likely to connect. Specifically, Katz counts the number of all paths between two nodes, emphasizing shorter paths by penalizing longer ones with a factor. SimRank assumes that two nodes are similar if they are linked to similar nodes, and PPR produces a ranking personalized to a particular node based on the random walk. (3) **Feature proximity heuristics** is available when node attributes are available, incorporating the side information about individual nodes. (Nickel et al., 2014; Zhao et al., 2017) combines graph structure with latent features and explicit features for better performance.

**Graph Neural Networks** Graph Neural Networks (GNNs) have emerged as a powerful technique in Deep Learning, specifically designed for graph-structured data. They address the limitations of traditional neural networks in dealing with irregular data structures. GNNs learn node representations by aggregating neighborhood and transforming features recursively.

Graph Neural Networks (Kipf & Welling, 2017; 2016; Hamilton et al., 2017; Velickovic et al., 2017) aim to learn high-quality features from data instead of predefined heuristics developed with domain knowledge. Despite satisfying results in many tasks via utilizing structural information, (Zhang & Chen, 2018; Zhang et al., 2021; Liang et al., 2022) find that vanilla GNNs are still sub-optimal suffering from disability on capturing important pairwise patterns for Link Prediction, e.g., Common Neighborhood. Graph Neural Networks for Link Prediction (GNN4LP) (Zhang & Chen, 2018; Zhang et al., 2021; Yun et al., 2021; Wang et al., 2022; 2023; Chamberlain et al., 2023; Zhu et al., 2021b) are then proposed which incorporates different inductive bias to capturing more pairwise information. SEAL (Zhang & Chen, 2018), Neo-GNN (Yun et al., 2021) and NCNC (Wang et al., 2023) involve more neighbor-overlapping knowledge. BUDDY (Chamberlain et al., 2023), and NBFNet (Zhu et al., 2021b) exploit the higher order structural information, e.g., number of paths between nodes. Nonetheless, advanced GNN4LP models with better effectiveness usually have more complicated model designs, leading to the efficiency issue. (Yin et al., 2022; 2023; Wu et al., 2021; Zhu et al., 2022; Kong et al., 2022) are then proposed to improve efficiency via approximating methods, e.g., hashing algorithm, A* algorithm, and random walk approximations.

The node embedding models (Yang et al., 2015; Grover & Leskovec, 2016) are graph learning algorithms used to transform the nodes of a graph into low-dimensional vectors, capturing the network topology and the node feature properties. This approach learns representations through data-driven algorithms, enabling effective capturing of complex patterns within the graph structure and node features for link prediction. For instance, Node2Vec (Grover & Leskovec, 2016) captures

both local and global structural proximity while TADW (Grover & Leskovec, 2016) [3] captures both local structural proximity and feature proximity.

Notably, our paper provides a comprehensive analysis with the latent space model to understand the important data factors for the link prediction task and how different link prediction algorithms work. The analysis can be used to understand all the above algorithms including graph embedding algorithms, GNNs, and heuritics in a holistic manner.

**Principles in Link Prediction & Network Analysis** Over the years, various theories and expertise knowledge have emerged to understand and predict links in networks. We typically review the foundational theories shaping the landscape of link prediction and network analysis. Moreover, we provide a detailed comparison with particular works.

Homophily (Khanam et al., 2020) is a long-standing principle in social network analysis, stemming from the shared beliefs and thoughts of individuals. It suggests that people with aligned perspectives are likely to connect with one another, despite potential differences in their social position. (Murase et al., 2019; Evtushenko & Kleinberg, 2021; Khanam et al., 2020; Currarini et al., 2009; McPherson et al., 2001) provide further study illustrating how homophily induced different phenomenons in social networks.

Tradic closure (Huang et al., 2015) is another fundamental concept in social network analysis, stemming from that friends of friends become friends themselves. It suggests that two individuals, who have a mutual friend, become connected themselves. (Dong et al., 2017; Rossi et al., 2020; Huang et al., 2018; Opsahl, 2013) provide further study illustrating how homophily induced different phenomenons in social networks. (Sarkar et al., 2011) theoretically proves the effectiveness of heuristics algorithms. The key differences between our work and (Sarkar et al., 2011) are as follow: 1. (Sarkar et al., 2011) assumes nodes are with the same degree in most cases, while our model can adapt to graphs with all kinds of degree distribution. 2. (Sarkar et al., 2011) only focuses on the structural perspective while our model considers both effects from feature and structural 3. (Sarkar et al., 2011) focuses on the deterministic model while the non-deterministic one is largely ignored.

More recently, (Murase et al., 2019; Asikainen et al., 2020; Abebe et al., 2022) focus on investigating the interplay between the role of the tradic closure and homophily. (Asikainen et al., 2020) first demonstrate that triadic closure intensifies the impacts of homophily. Nonetheless, (Abebe et al., 2022) points out that (Asikainen et al., 2020) builds on the existence of sufficient tradic closure rather than considering the tradic closure and homophily simultaneously. With a modest modification, (Abebe et al., 2022) finds that tradic closure can introduce individuals to those unlike themselves, consequently reducing segregation. The key differences between our work and (Abebe et al., 2022) are as follows: (1) (Abebe et al., 2022) typically focuses on alleviating the effects of segregation (Tóth et al., 2021) via the tradic closure rather than the link prediction task in our work. (2) (Abebe et al., 2022) focuses on the typical graph model in social science, while our work aims to understand link prediction in various domains.

Despite those principles focusing on network analysis, more theories and principles are proposed revolving around deep learning models, especially for GNN4LP models. (Zhang & Chen, 2018) proposes the $\gamma$-decaying heuristic theory indicating that subgraph GNN can capture sufficient structural information for link prediction. (Zhang et al., 2021) proposes the labeling trick theory indicating how to identify the most structural node representation. (Zhou et al., 2022) revolves around the size stability of inductive OOD link prediction problem from a causal perspective.

## B   DETAILS ON HEURISTIC ALGORITHMS

We then explain more implementation details on those heuristics as follows:

**Katz.** $\lambda < 1$ is a damping factor, indicating the importance of the higher-order information. We set $\lambda = 0.1$ in our implementation. $|\text{paths}^{\langle l \rangle}(i,j)|$ counts the number of length-$l$ paths between $i$ and $j$. We compute the Katz index (Katz, 1953) by approximating the closed-form solution $S = (I - \lambda A)^{-1} - I$ with $\lambda A + \lambda^2 A^2 + \cdots + \lambda^n A^n$, where $S$ is the score matrix. We utilize $\lambda = 0.05$ and $n = 3$ in our implementation.

**PPR.** $[\pi_i]_j$ is the stationary distribution probability of $j$ under the random walk from $i$ with restart, see more details in Brin & Page (2012). We adapt 1e-3 as the stop criterion.

Table 2: Popular heuristics for link prediction, where $\Gamma(i)$ denotes the neighbor set of vertex $i$.

| Name | Formula | Factor |
|------|---------|--------|
| common neighbors (CN) | $\|\Gamma(i) \cap \Gamma(j)\|$ | LSP |
| Adamic-Adar (AA) | $\sum_{k \in \Gamma(i) \cap \Gamma(j)} \frac{1}{\log \|\Gamma(k)\|}$ | LSP |
| resource allocation (RA) | $\sum_{k \in \Gamma(j) \cap \Gamma(i)} \frac{1}{\|\Gamma(k)\|}$ | LSP |
| Katz | $\sum_{l=1}^{\infty} \lambda^l \|\text{paths}^{\langle l \rangle}(i,j)\|$ | GSP |
| Personal PageRank (PPR) | $[\pi_i]_j + [\pi_j]_i$ | GSP |
| SimRank | $\gamma \frac{\sum_{a \in \Gamma(i)} \sum_{b \in \Gamma(j)} \text{score}(i,j)}{\|\Gamma(i)\| \cdot \|\Gamma(j)\|}$ | GSP |
| Feature Homophily (FH) | $\text{dis}(x_i, x_j)$ | FP |

**SimRank.** We utilize the efficient framework (Zhu et al., 2021a) with the LocalPush SimRank algorithm (Wang et al., 2019b). Code is available at https://github.com/UISim2020/UISim2020.

**FH.** It estimates the node feature similarity between a pair of nodes. Different feature distance metrics can be utilized for estimation. Typically, we include two feature distance metrics: cosine distance and Euclidean distance.

**Double-Radius Node Labeling.** It aims to extract the position information based on the source and target nodes $i$ and $j$. It can be calculated as: $f_l(i) = 1 + \min(d_x, d_y) + (d/2)[(d/2) + (d\%2) - 1]$ where $d_x = d(i,x)$, $d_y = d(y,i)$. $d(i,j)$ is the node with a distance $i$ to the source node and a distance $j$ to the target node.

**Heuristic selection** . In section 3, we majorly focus on one default heuristic for each factor, which are CN for LSP, Katz for GSP, and FH for FP. Moreover, for experiments in Section 3.1, we make a comparison between two heuristics from the same factors. The details for the second heuristic are RA for LSP, PPR for GSP. For FP factor, the default one is with cosine distance while the second one is with Euclidean distance.

## C  THEORETICAL ANALYSIS ON THE GRAPH LATENT SPACE MODEL

In Section 3.2, we introduce the latent space model and theoretically verify the effectiveness of heuristic algorithms derived from three factors: local structural proximity (LSP), global structural proximity (GSP), and feature proximity (FP). In this section, we provide all the proof details in Section 3.2 and 3.3 on the signifiance of the data factors and their underlying relationship. In particular, the following proof is inspired by the latent model proposed in (Sarkar et al., 2011). The main differences between our model and (Sarkar et al., 2011) are two-fold with better alignment to the real-world scenario. (1) Our model can easily extend to graphs under arbitrary degree distributions in (Sarkar et al., 2011) rather than only considering regular graphs with the same node degree. (2) Our model adds node features into consideration rather than only considering unfeatured graphs in (Sarkar et al., 2011). Moreover, we provide a deep analysis of the interplay between feature and structure proximity.

### C.1  EXTENDED LATENT SPACE MODEL FOR LINK PREDICTION

To ease the analysis, we utilize the latent space model as the data assumption showing as follows:

**Definition 1** (Latent space model for link prediction). *The generated nodes are uniformly distributed in a $D$ dimensional Euclidean space. Each node possesses a subordinate radius $r$ and corresponding volume $V(r)$. For any nodes $i, j$, The probability of link formation $P(i \sim j)$ is determined by the radius $(r_i, r_j)$ and distance $d_{ij}$ of the nodes at its ends.*

*1) $V(r) = V(1)r^D$, where $V(r)$ is the volume of a radius $r$ and $V(1)$ is the volume of a unit radius hypersphere.*

*2) $\text{Deg}(i) = NV(r_i)$, where $\text{Deg}(i)$ is degree of node $i$ and $N$ is the total number of nodes.*

3) $P(i \sim j | d_{ij}) = 1/(1 + e^{\alpha(d_{ij} - \max\{r_i, r_j\})})$, where $P(i \sim j | d_{ij})$ depicts the probability of forming a unidirectional link between $i$ and $j$ ($i \sim j$), $\alpha > 0$ controls the sharpness of the function and $r$ determines the threshold.

In order to normalize the probabilities, we assume that all points lie inside a unit volume hypersphere in $D$ dimensions. The maximum $r_{MAX}$ satisfies $V(r_{MAX}) = V(1) r_{MAX}^D = 1$, i.e., $r_{MAX} = (\frac{1}{V(1)})^{1/D}$. For any node $i,j$ in graph, we define the volume of intersection of two balls of $V(r_i)$ and $V(r_j)$ as $A(r_i, r_j, d_{ij})$, which can be bounded using hypersphere as:

$$\left( \frac{r_i + r_j - d_{ij}}{2} \right)^D \leq \frac{A(r_i, r_i, d_{ij})}{V(1)} \leq \left( r_{ij}^{max} - \left( \frac{d_{ij}}{2} \right)^2 \right)^{D/2} \tag{2}$$

where $r_{ij}^{max} = max\{r_i, r_j\}$ and Eq.(2) above bridges $A(r_i, r_j, d_{ij})$ and $d_{ij}$ through the volume of hypersphere.

The latent space model above is well-fitted for link prediction because the distance $d_{ij}$ in the model portrays the probability of forming a link between node $i$ and $j$, and for a given node, the most likely node it would connect to is the non-neighbor at the smallest distance. In addition, $\text{Deg}(i) = NV(r_i)$ describes the sum of the first-order link neighbors of the node. Our latent space model can be applied to the dataset where nodes are distributed independently in some latent metric space; given the positions links are independent of each other.

## C.2 The effectiveness of local structural proximity (LSP)

**Proposition 1** (latent space distance bound with CNs). *For any $\delta > 0$, with probability at least $1 - 2\delta$, we have $d_{ij} \leq 2\sqrt{r_{ij}^{max} - \left( \frac{\eta_{ij}/N - \epsilon}{V(1)} \right)^{2/D}}$, where $\eta_{ij}$ is the number of common neighbors between nodes $i$ and $j$, $r_{ij}^{max} = max\{r_i, r_j\}$, and $V(1)$ is the volume of a unit radius hypersphere in $D$ dimensional space. $\epsilon$ is a term independent of $\eta_{ij}$. It vanishes as the number of nodes $N$ grows.*

Note $\mathcal{N}(i)$ as the set of neighbors of node $i$ and let $Y_k$ be a random variable depending on the position of point $k$, which is 1 if $k \in \mathcal{N}(i) \cap \mathcal{N}(j)$, and 0 otherwise. Therefore, we have

$$E[Y_k | d_{ij}] = P(i \sim k \sim j | d_{ij}) = \int_{d_{ik}, d_{jk}} P(i \sim k | d_{ik}) P(j \sim k | d_{jk}) P(d_{ik}, d_{jk} | d_{ij}) d(d_{ij})$$
$$= A(r_i, r_j, d_{ij}) \tag{3}$$

For the sake of brevity, we denote $E[Y_k | d_{ij}]$ as $E[Y_k]$. It can be easy observed that $\sum_{k=0}^{N} Y_k = \eta_{ij}$, i.e., the common neighbour number (CN) of node $i$ and $j$, denoted as $\eta_{ij}$ here. Through empirical Bernstein bounds (Maurer & Pontil, 2009), we have

$$P\left[ \left| \sum_k^N Y_k/N - E[Y_k] \right| \geq \sqrt{\frac{2 \text{var}_N(Y) \log 2/\delta}{N}} + \frac{7 \log 2/\delta}{3(N-1)} \right] \leq 2\delta \tag{4}$$

where $\text{var}_N(Y) = \frac{\eta_{ij}(1 - \eta_{ij}/N)}{N-1}$ is the sample variance of $Y$. Setting $\epsilon = \sqrt{\frac{2 \text{var}_N(Y) \log 2/\delta}{N}} + \frac{7 \log 2/\delta}{3(N-1)}$

$$P\left[ \frac{\eta_{ij}}{N} - \epsilon \leq A(r_i, r_j, d_{ij}) \leq \frac{\eta_{ij}}{N} + \epsilon \right] \geq 1 - 2\delta \tag{5}$$

Combining Eq. (2) and equation above, we can get the bounds of $d_{ij}$ as:

$$\begin{cases} \left( \frac{r_i + r_j - d_{ij}}{2} \right)^D V(1) \leq \frac{\eta_{ij}}{N} + \epsilon \\ \left( r_{ij}^{max} - \left( \frac{d_{ij}}{2} \right)^2 \right)^{D/2} V(1) \geq \frac{\eta_{ij}}{N} - \epsilon \end{cases} \tag{6}$$

i.e.,

$$r_i + r_j - 2\left(\frac{\eta_{ij}/N + \epsilon}{V(1)}\right)^{1/D} \leq d_{ij} \leq 2\sqrt{r_{ij}^{max} - \left(\frac{\eta_{ij}/N - \epsilon}{V(1)}\right)^{2/D}} \tag{7}$$

The R.H.S of the above equation, i.e., the upper bound of $d_{ij}$ decreases as $\eta_{ij}$ increases. In other words, the greater CN number $\eta_{ij}$ of two nodes $i,j$ will cause the upper bound on the distance $d_{ij}$ between them to decrease more, thus indicating the effectiveness of local structural proximity.

Notably, the above theoretical analysis can be easily extended to many other local heuristics, e.g., RA, AA (Zhou et al., 2009), which can be viewed as a weighted version for common neighbors. The key to extending proof to those heuristics is to change the definition of $Y_k$ mainly distinguished from CN by their weight design, can also be proven the existence of similar conclusions. Details show in section C.5

### C.3   THE EFFECTIVENESS OF GLOBAL STRUCTURAL PROXIMITY (GSP)

**Proposition 2** (latent space distance bound with the number of paths). *For any $\delta > 0$, with probability at least $1 - 2\delta$, we have $d_{ij} \leq \sum_{n=0}^{M-2} r_n + 2\sqrt{r_M^{\max} - \left(\frac{\eta_\ell(i,j) - b(N,\delta)}{c(N,\delta,\ell)}\right)^{\frac{2}{D(\ell-1)}}}$ , where $\eta_\ell(i,j)$ is the number of paths of length $\ell$ between $i$ and $j$ in $D$ dimensional Euclidean space. $M \in \{1, \cdots, \ell - 1\}$ is the set of intermediate nodes.*

In this session, we analyze how $d_{ij}$ can be upper-bounded through the number of simple $\ell$-hop paths.

**Definition 1.**  *(simple path and set) Given node $i$, $j$ in graph $G(V, E)$, define a simple path of length $\ell$ between $i$, $j$ as $path(i, k_1, k_2, \cdots, k_{\ell-2}, j)$ such that $i \sim k_1 \sim k_2 \sim \ldots k_{\ell-2} \sim j$ and $S_\ell(i, j)$ as the set of all possible $path(i, k_1, k_2, \cdots, k_{\ell-2}, j)$, $\{k_1, k_2, \cdot, k_{l-2}\} \in V$*

Let $Y(i, k_1, k_2, \cdots, k_{\ell-2}, j)$ be a random variable which is 1 if $(i, k_1, k_2, \cdots, k_{\ell-2}, j) \in S_\ell(i, j)$ and 0 else. We denote $\eta_\ell(i, j)$ as the number of paths of length $\ell$ between $i$ and $j$.

$$\eta_\ell(i, j) = \sum_{k_1, \ldots k_{\ell-2} \in \mathcal{S}^{(\ell-2)}} Y(i, k_1, \ldots, k_{\ell-2}, j \mid d_{ij}) \tag{8}$$

Our goal is to infer bounds on $d_{ij}$ given the observed number of $\eta_\ell(i, j)$. We first bound the maximum degree $\Delta$ as follows.

**Lemma 1.**  $\Delta < N\left(1 + \sqrt{-2\ln\delta/N}\right)$ *with probability at least $1 - \delta$.*

**Proof:** The degree $\text{Deg}(k)$ of any node $k$ is a binomial random variable with expectation $E[\text{Deg}(k)] = NV(r_k)$, where $V(r_k)$ is the volume of a hypersphere of radius $r_k$. Thus, using the Chernoff bound, $\text{Deg}(k) < NV(r_k)\left(1 + \sqrt{\frac{-2\ln\delta}{NV(r_k)}}\right)$ holds with probability at least $1 - \delta$. Applying the union bound on all nodes yields the desired proposition, i.e., $\Delta < NV(r_{MAX})\left(1 + \sqrt{\frac{-2\ln\delta}{NV(r_{MAX})}}\right) = N\left(1 + \sqrt{-2\ln\delta/N}\right)$.

Next, we use $\Delta$ to bound the maximum possible value of $\eta_\ell(i, j)$.

**Lemma 2.**  *For any graph with maximum degree $\Delta$, we have: $\eta_\ell(i, j) \leq \Delta^{\ell-1}$.*

**Proof:** This can be proved using a simple inductive argument. If the graph is represented by adjacency matrix $M$, then the number of length $\ell$ paths between $i$ and $j$ is given by $M^\ell(i, j)$. Trivially $M_{ij}^2$ can be at most $\Delta$. This happens when both $i$ and $j$ have degree $\Delta$, and their neighbors form a perfect matching. Assuming this is true for all $m < \ell$, we have: $M^\ell(i, j) = \sum_p M(i, p)M^{\ell-1}(p, j) \leq \Delta^{\ell-2}\sum_p M(i, p) \leq \Delta^{\ell-1}$

**Lemma 3.**  *For $\ell < \Delta, \left|\eta_\ell(i, j \mid X_1, \ldots, X_p, \ldots, X_N) - \eta_\ell\left(i, j \mid X_1, \ldots, \tilde{X}_p, \ldots X_N\right)\right| \leq (\ell - 1) \cdot \Delta^{\ell-2}$.*

**Proof:** The largest change in $\eta_\ell(\cdot)$ occurs when node $p$ was originally unconnected to any other node, and is moved to a position where it can maximally add to the number of $\ell$-hop paths between

$i$ and $j$ (or vice versa). Consider all paths where $p$ is $m$ hops from $i$ (and hence $\ell - m$ hops from $j$. From Lemma 2, the number of such paths can be at most $\Delta^{m-1} \cdot \Delta^{\ell-m-1} = \Delta^{\ell-2}$. Since $m \in \{1, \ldots, \ell - 1\}$, the maximum change is $(\ell - 1) \cdot \Delta^{\ell-2}$.

Define $P_\ell(i, j)$ as the probability of observing an $\ell$-hop path between points $i$ and $j$. Next, we compute the expected number of $\ell$-hop paths.

**Theorem 1.** $E\left[\eta_\ell(i, j)\right] \leq \Delta^{\ell-1} \prod_{p=1}^{\ell-1} A\left(r, p \times r, (d_{ij} - (\ell - p - 1)r)_+\right)$, where $x_+$ is defined as $\max(x, 0)$.

**Proof:** Consider an $\ell$-hop path between $i, j$, for clarity of notation, let us denote the distances $d_{i,k_1}, d_{k_1,k_2}$ etc. by $a_1, a_2$, up to $a_{\ell-1}$ and radius $r_i, r_{k_1}, \cdots, r_j$ by $r_0, r_1, \cdots, r_{\ell-1}$. We also denote the distances $d_{jk_1}, d_{jk_2}$, etc. by $d_1, d_2, \cdots, d_{\ell-1}$. Note $r'_j = \max(r_{j-1}, r_j), j \in \{1, 2, \cdots, \ell - 1\}$ From the triangle inequality, $d_{\ell-2} \leq a_{\ell-1} + a_\ell \leq r_{\ell-1} + r_\ell$, and by induction, $d_k \leq \sum_{m=k+1}^\ell r_m$. Similarly, $d_1 \geq (d_{ij} - a_1)_+ \geq (d_{ij} - r_i)_+$, and by induction, $d_k \geq \left(d_{ij} - \sum_{n=0}^{k-1} r_n\right)_+$.

$$
\begin{aligned}
P_\ell(i, j) &= P\left(i \sim k_1 \sim \ldots \sim k_{\ell-1} \sim j \mid d_{ij}\right) \\
&= P\left(a_1 \leq r'_1 \cap \ldots \cap a_\ell \leq r'_{\ell-1} \mid d_{ij}\right) \\
&= \int_{d_1, \ldots, d_{\ell-2}} P\left(a_1 \leq r'_1, \ldots, a_{\ell-1} \leq r'_{\ell-1}, d_1, \ldots, d_{\ell-2} \mid d_{ij}\right) \\
&= \int_{d_{\ell-2}=(d_{ij}-\sum_{n=0}^{\ell-3} r_n)_+}^{r_{\ell-1}+r_\ell} \cdots \int_{d_1=(d_{ij}-r_0)_+}^{\sum_{m=2}^\ell r_m} P\left(a_1 \leq r'_1, d_1 \mid d_{ij}\right) \ldots P\left(a_{\ell-1} \leq r'_{\ell-1}, a_\ell \leq r'_\ell \mid d_{\ell-2}\right) \\
&\leq A\left(r'_1, \sum_{m=2}^\ell r_m, d_{ij}\right) \times A\left(r'_2, \sum_{m=3}^\ell r_m, (d_{ij} - r_0)_+\right) \times \ldots \times A\left(r'_{\ell-1}, r_\ell, (d_{ij} - \sum_{n=0}^{\ell-3} r_n)_+\right) \\
&\leq \prod_{p=1}^{\ell-1} A\left(r'_p, \sum_{m=p+1}^\ell r_m, \left(d_{ij} - \sum_{n=0}^{p-2} r_n\right)_+\right)
\end{aligned}
$$

(9)

Since there can be at most $\Delta^{\ell-1}$ possible paths (from Lemma 2), the theorem statement follows.

**Theorem 2.** $\eta_\ell(i, j) \leq \left[\prod_{p=1}^{\ell-1} A\left(r'_p, \sum_{m=p+1}^\ell r_m, \left(d_{ij} - \sum_{n=0}^{p-2} r_n\right)_+\right) + \frac{(\ell-1)\sqrt{\frac{\ln(1/\delta)}{2}}}{\sqrt{N}\left(1+\sqrt{\frac{-2\ln\delta}{N}}\right)}\right] \cdot$ $\left(N + \sqrt{-2N\ln\delta}\right)^{\ell-1}$ with probability at least $(1 - 2\delta)$.

**Proof:** From McDiarmid's inequality (McDiarmid et al., 1989), we have:

$$
\eta_\ell(i, j) \leq E\left[\eta_\ell(i, j)\right] + (\ell - 1)\Delta^{\ell-2}\sqrt{\frac{N\ln(1/\delta)}{2}} \tag{10}
$$

$$
\leq E\left[\eta_\ell(i, j)\right] + \Delta^{\ell-1}\sqrt{\frac{N\ln(1/\delta)}{2}} \tag{11}
$$

$$
\leq \Delta^{\ell-1}\left[\prod_{p=1}^{\ell-1} A\left(r'_p, \sum_{m=p+1}^\ell r_m, \left(d_{ij} - \sum_{n=0}^{p-2} r_n\right)_+\right) + \sqrt{\frac{N\ln(1/\delta)}{2}}\right] \tag{12}
$$

$$
\leq \left[\prod_{p=1}^{\ell-1} A\left(r'_p, \sum_{m=p+1}^\ell r_m, \left(d_{ij} - \sum_{n=0}^{p-2} r_n\right)_+\right) + \sqrt{\frac{N\ln(1/\delta)}{2}}\right] \cdot \left(N + \sqrt{-2N\ln\delta}\right)^{\ell-1} \tag{13}
$$

The inequality above can be rewritten as:

$$
\eta_\ell(i, j) \leq c(N, \delta, \ell) \prod_{p=1}^{\ell-1} A\left(r'_p, \sum_{m=p+1}^\ell r_m, \left(d_{ij} - \sum_{n=0}^{p-2} r_n\right)_+\right) + b(N, \delta) \tag{14}
$$

where $c(N, \delta, \ell)$ and $b(N, \delta)$ contain coefficients that do not vary with $d_{ij}$. Note $r_p^{max} = \max\{r'_p, \sum_{m=p+1}^{\ell} r_m\}$, and $d_{ij}$ can be upper-bounded through the number of simple $\ell$-hop paths $\eta_\ell(i, j)$ with Eq. (2).

$$\eta_\ell(i, j) \leq c(N, \delta, \ell) \prod_{p=1}^{\ell-1} \left( r_p^{max} - \left( \frac{\left( d_{ij} - \sum_{n=0}^{p-2} r_n \right)_+}{2} \right)^2 \right)^{D/2} + b(N, \delta)$$

$$= c(N, \delta, \ell) \left( \prod_{p=1}^{\ell-1} \left[ r_p^{max} - \left( \frac{d_{ij} - \sum_{n=0}^{p-2} r_n}{2} \right)^2 \right] \right)^{D/2} + b(N, \delta)$$

$$\leq c(N, \delta, \ell) \left( \prod_{p=1}^{\ell-1} \left[ r_M^{max} - \left( \frac{d_{ij} - \sum_{n=0}^{M-2} r_n}{2} \right)^2 \right] \right)^{D/2} + b(N, \delta) \quad \exists M \in \{1, \cdots, \ell-1\}$$

$$\leq c(N, \delta, \ell) \left( r_M^{max} - \left( \frac{d_{ij} - \sum_{n=0}^{M-2} r_n}{2} \right)^2 \right)^{D(\ell-1)/2} + b(N, \delta)$$

i.e.,

$$d_{ij} \leq \sum_{n=0}^{M-2} r_n + 2\sqrt{r_M^{max} - \left( \frac{\eta_\ell(i, j) - b(N, \delta)}{c(N, \delta, \ell)} \right)^{\frac{2}{D(\ell-1)}}} \tag{15}$$

Since b is ignorable, we can observe the upper bound decreases as $\eta_\ell(i, j)$ increases which implies the effectiveness of GSP as well.

**The relationship between global structural proximity (GSP) and local structural proximity (LSP) :**

The detailed proof of the relationship between global and local structural proximity is shown as follows. The key is to view the CN algorithm as the count on the number of paths $\eta_\ell(i, j)$ with length $\ell = 2$.

**Lemma 4** (latent space distance bound with local and global structural proximity). *For any $\delta > 0$, with probability at least $1 - 2\delta$, we have $d_{ij} \leq \sum_{n=0}^{M-2} r_n + 2\sqrt{r_M^{max} - \left( \sqrt{\frac{N \ln(1/\delta)}{2}} - 1 \right)^{\frac{2}{D(\ell-1)}}}$, where $\sum_{n=0}^{M-2} r_n$, $r_M^{max}$ serve as independent variables that do not change with $\ell$.*

**Proof:** Replace $c$ and $b$ in Eq. (15), we can obeserve that

$$d_{ij} \leq \sum_{n=0}^{M-2} r_n + 2\sqrt{r_M^{max} - \left( \sqrt{\frac{N \ln(1/\delta)}{2}} - \frac{\eta_\ell(i, j)}{\left( N + \sqrt{-2N \ln \delta} \right)^{\ell-1}} \right)^{\frac{2}{D(\ell-1)}}} \tag{16}$$

$$\leq \sum_{n=0}^{M-2} r_n + 2\sqrt{r_M^{max} - \left( \sqrt{\frac{N \ln(1/\delta)}{2}} - \left( \frac{\Delta}{N + \sqrt{-2N \ln \delta}} \right)^{\ell-1} \right)^{\frac{2}{D(\ell-1)}}} \tag{17}$$

$$\leq \sum_{n=0}^{M-2} r_n + 2\sqrt{r_M^{max} - \left( \sqrt{\frac{N \ln(1/\delta)}{2}} - \left( \frac{1}{1 + \sqrt{\frac{-2 \ln \delta}{N}}} \right)^{\ell-1} \right)^{\frac{2}{D(\ell-1)}}} \tag{18}$$

$$\leq \sum_{n=0}^{M-2} r_n + 2\sqrt{r_M^{max} - \left( \sqrt{\frac{N \ln(1/\delta)}{2}} - 1 \right)^{\frac{2}{D(\ell-1)}}} \tag{19}$$

Since $\ell$ in $\left(\sqrt{\frac{N\ln(1/\delta)}{2}}-1\right)^{\frac{2}{D(\ell-1)}}$ acts as an exponential coefficient, the upper bound of $d_{ij}$ grows at a surprisingly fast rate as $\ell$ increases (i.e. the order of the structure's proximity goes higher), which makes the effectiveness of structure proximity much weaker.

## C.4 THE EFFECTIVENESS OF FEATURE PROXIMITY (FP) & RELATIONSHIP WITH FEATURE PROXIMITY

**Proposition 3** (latent space distance bound with feature proximity). *For any $\delta > 0$, with probability at least $1 - 2\delta$, we have $d_{ij} \leq 2\sqrt{r_{ij}^{max} - \left(\frac{\beta_{ij}(1-A(r_i,r_j,d_{ij}))+A(r_i,r_j,d_{ij})}{V(1)}\right)^{2/D}}$, where $\beta_{ij}$ measures feature proximity between $i$ and $j$, $r_{ij}^{max} = max\{r_i, r_j\}$ and $V(1)$ is the volume of a unit radius hypersphere in $D$ dimensional Euclidean space. $A(r_i, r_j, d_{ij})$ is the volume of intersection of two balls of $V(r_i)$ and $V(r_j)$ in latent space, corresponding to the expectation of common neighbors.*

For graphs with node features, feature proximity (FP) is also often used as a similarity measure. In this section, we briefly analyze the effect of FP on $d_{ij}$. Since FP is not directly related to the spatial concepts of the individuals in the model (e.g., radius $r$, distance $d$), we can think of it as a noise parameter that affects the probability of a connecting edge of two nodes. According to the original deterministic model, $i \sim j$ iff $d_{ij} < max\{r_i, r_j\}$, it relies almost exclusively on structural proximity. The introduction of the FP can serve as a noise parameter $\beta_{ij}$ to extend the deterministic model further into a non-deterministic model, i.e. $i \sim j$ with probability $\beta_{ij} \in (0, 1)$ (if $d_{ij} > max\{r_i, r_j\}$), or with probability $1 - \beta_{ij}$ (otherwise). Now, the probability of having a common neighbor between node $i$, $j$ will be $A_{\beta_{ij}}(r_i, r_j, d_{ij}) = \beta_{ij} + (1 - \beta_{ij})A(r_i, r_j, d_{ij})$. Substituting $A_{\beta_{ij}}(r_i, r_j, d_{ij})$ for $A(r_i, r_j, d_{ij})$ in Eq. (2), we have:

$$\left(\frac{r_i + r_j - d_{ij}}{2}\right)^D \leq \frac{A_\beta(r_i, r_j, d_{ij})}{V(1)} \leq \left(r_{ij}^{max} - \left(\frac{d_{ij}}{2}\right)^2\right)^{D/2} \tag{20}$$

The upper bound of $d_{ij}$ is

$$d_{ij} \leq 2\sqrt{r_{ij}^{max} - \left(\frac{\beta_{ij}(1 - A(r_i, r_j, d_{ij})) + A(r_i, r_j, d_{ij})}{V(1)}\right)^{2/D}} \tag{21}$$

An increase in FP value $\beta_{ij}$ can reduce the upper bound of $d_{ij}$, which reveals the effectiveness of FP. Since $A(r_i, r_j, d_{ij}) \in [0, 1)$ and $\beta \in (0, 1)$.

**The relationship between local structural proximity (LSP) and feature proximity (FP):**

**Lemma 5** (Incompatibility between LSP and FP factors). *For any $\delta > 0$, with probability at least $1 - 2\delta$, we have $\eta_{ij} = \frac{c'}{1-\beta_{ij}} + N(1 + \epsilon)$, where $\eta_{ij}$ and $\beta_{ij}$ are the number of common neighbor nodes and feature proximity between nodes $i$ and $j$. $c' < 0$ is an independent variable that does not change with $\beta_{ij}$ and $\eta_{ij}$. $\eta_{ij}$ is negatively correlated with $\beta_{ij}$.*

**Proof :** Combining Eq. (5) and Eq. (21), we have for any $\delta > 0$, with probability at least $1 - 2\delta$,

$$d_{ij} \leq U_{d_{ij}} = 2\sqrt{r_{ij}^{max} - \left(\frac{\beta_{ij} + (1 - \beta_{ij})(\frac{\eta_{ij}}{N} - \epsilon)}{V(1)}\right)^{2/D}} \tag{22}$$

where $\epsilon = \sqrt{\frac{2\,\text{var}_N(Y)\log 2/\delta}{N}} + \frac{7\log 2/\delta}{3(N-1)}$, $\text{var}_N(Y) = \frac{\eta_{ij}(1-\eta_{ij}/N)}{N-1}$ is the sample variance of $Y$, and $\eta_{ij}$ represents the common neighbour number between node $i$ and $j$. Eq. (22) can be rewritten as:

$$\eta_{ij} = \frac{c'(U_{d_{ij}}, r_{ij}^{max}, N)}{1 - \beta_{ij}} + N(1 + \epsilon) \tag{23}$$

where

$$c'(U_{d_{ij}}, r_{ij}^{max}, N) = NV(1)(r_{ij}^{max} - (\frac{U_{d_{ij}}}{2})^2)^{D/2} - N \tag{24}$$

$$\leq NV(1)(r_{ij}^{max} - (\frac{d_{ij}}{2})^2)^{D/2} - N \tag{25}$$

$$< NV(1)\frac{1}{V(1)} - N = 0 \tag{26}$$

Since $\beta \in (0, 1)$, we can learn from Eq. (23) that numerically, $\eta_{ij}$ decreases as $\beta_{ij}$ increases, and vice-versa. It indicates that when the ground truth $d_{ij}$ is determined, we can learn from Eq. (23) that there is a conflict between LSP $\eta_{ij}$ and FP $\beta_{ij}$: these two cannot be increased at the same time.

**The relationship between global structural proximity (GSH) and feature proximity (FP)**

After the introduction of feature proximity into the model, we can rewrite Eq. (14) as:

$$\eta_\ell(i, j) \leq c(N, \delta, \ell) \prod_{p=1}^{\ell-1} A_\beta \left( r'_p, \sum_{m=p+1}^{\ell} r_m, \left( d_{ij} - \sum_{n=0}^{p-2} r_n \right)_+ \right) + b(N, \delta)$$

$$= c(N, \delta, \ell) \prod_{p=1}^{\ell-1} \left[ \beta_{ij} + (1 - \beta_{ij}) A \left( r'_p, \sum_{m=p+1}^{\ell} r_m, \left( d_{ij} - \sum_{n=0}^{p-2} r_n \right)_+ \right) \right] + b(N, \delta)$$

$$\leq c(N, \delta, \ell) \left( \prod_{p=1}^{\ell-1} \left[ \beta_{ij} + (1 - \beta_{ij}) \left( r_p^{max} - \left( \frac{d_{ij} - \sum_{n=0}^{p-2} r_n}{2} \right)^2 \right)^{D/2} \right] \right) + b(N, \delta)$$

$$\leq c(N, \delta, \ell) \left( \left[ \beta_{ij} + (1 - \beta_{ij}) \left( r_M^{max} - \left( \frac{d_{ij} - \sum_{n=0}^{M-2} r_n}{2} \right)^2 \right)^{D/2} \right]^{\ell-1} \right) + b(N, \delta) \quad \exists M \in \{1, \cdots, \ell-1\}$$

$$\tag{27}$$

i.e.

$$d_{ij} \leq U_{d_{ij}} = \sum_{n=0}^{M-2} r_n + 2 \sqrt{r_M^{max} - \left( \frac{\left( \frac{\eta_\ell(i,j) - b(N,\delta)}{c(N,\delta,\ell)} \right)^{1/(\ell-1)} - 1}{1 - \beta_{ij}} + 1 \right)^{2/D}} \tag{28}$$

i.e.,

$$\beta_{ij} = 1 + \frac{\left( \frac{\eta_\ell(i,j) - b(N,\delta)}{c(N,\delta,\ell)} \right)^{1/(\ell-1)} - 1}{1 - \left[ r_M^{max} - \left( \frac{U_{d_{ij}} - \sum_{n=0}^{M-2} r_n}{2} \right)^2 \right]^{D/2}} \tag{29}$$

Since $\left( \frac{\eta_\ell(i,j) - b(N,\delta)}{c(N,\delta,\ell)} \right) > 1$, We can learn from Eq. (23) that numerically, $\beta_\ell(i, j)$ decreases as $\eta_{ij}$ increases, and vice-versa. It indicates that there is a conflict between GP $\eta_{ij}$ and FP $\beta_{ij}$: these two cannot be increased at the same time.

### C.5 THEORETICAL ANALYSIS FOR OTHER LOCAL STRUCTRAL HEURISTICS

Except for common neighbor number(CN), there are still many local heuristics metrics, e.g. RA, and AA. In this session, we will analyze the relationship between $d_{ij}$ and them , which gives a broader account of the effectiveness of local structural proximity (LSP). We define RA and AA between node $i$ and $j$ as $\eta_{ij}^{RA} = \sum_{k \in \Gamma(x) \cap \Gamma(y)} \frac{1}{\text{Deg}(k)}$ and $\eta_{ij}^{AA} = \sum_{k \in \Gamma(x) \cap \Gamma(y)} \frac{1}{log(\text{Deg}(k))}$. Note $\min(\text{Deg}_{xy}) = \min_k \text{Deg}(k), k \in \Gamma(x) \cap \Gamma(y)$ and $\max(\text{Deg}_{xy}) = \max_k \text{Deg}(k), k \in \Gamma(x) \cap \Gamma(y)$.

For RA, let $Y_k$ be a random variable depending on the position of point $k$, which is $Y_k = \frac{1}{\text{Deg}(k)}$ if $k \in \mathcal{N}(i) \cap \mathcal{N}(j)$, and 0 otherwise. Therefore we have:

$$
\begin{aligned}
E\left[Y_k \mid d_{ij}\right] &= \int_k \frac{1}{\text{Deg}(k)} \cdot P\left(i \sim k \sim j \mid d_{ij}\right) \\
&= \int_{d_{ik}, d_{jk}} \frac{1}{\text{Deg}(k)} \cdot P\left(i \sim k \mid d_{ik}\right) P\left(j \sim k \mid d_{jk}\right) P\left(d_{ik}, d_{jk} \mid d_{ij}\right) d\left(d_{ij}\right) \quad (30) \\
&\leq \frac{1}{\min(\text{Deg}_{xy})} A(r_i, r_j, d_{ij})
\end{aligned}
$$

For the sake of brevity, we denote $E\left[Y_k \mid d_{ij}\right]$ as $E\left[Y_k\right]$. Similarly, we can obtain:

$$
\frac{1}{\max(\text{Deg}_{xy})} A(r_i, r_j, d_{ij}) \leq E\left[Y_k\right] \leq \frac{1}{\min(\text{Deg}_{xy})} A(r_i, r_j, d_{ij}) \quad (31)
$$

It can be easy observed that $\sum_{k=0}^{N} Y_k = \eta_{ij}^{RA}$, i.e., the RA number of node $i$ and $j$, denoted as $\eta_{ij}^{RA}$ here. Through empirical Bernstein bounds (Maurer & Pontil, 2009), we have:

$$
P\left[\left|\sum_k^N Y_k/N - E\left[Y_k\right]\right| \geq \sqrt{\frac{2\,\text{var}_N(Y)\log 2/\delta}{N}} + \frac{7\log 2/\delta}{3(N-1)}\right] \leq 2\delta \quad (32)
$$

where $\text{var}_N(Y) = \frac{\eta_{ij}^{RA}(1 - \eta_{ij}^{RA}/N)}{N-1}$ is the sample variance of $Y$. Setting $\epsilon = \sqrt{\frac{2\,\text{var}_N(Y)\log 2/\delta}{N}} + \frac{7\log 2/\delta}{3(N-1)}$.

$$
P\left[\frac{\eta_{ij}^{RA}}{N} - \epsilon \leq E[Y_k] \leq \frac{\eta_{ij}^{RA}}{N} + \epsilon\right] \geq 1 - 2\delta \quad (33)
$$

Combining Eq. (2) and equation above, we can get the bounds of $d_{ij}$ as:

$$
\begin{cases}
\dfrac{V(1)}{\max(\text{Deg}_{xy})} \left(\dfrac{r_i + r_j - d_{ij}}{2}\right)^D \leq \dfrac{\eta_{ij}^{RA}}{N} + \epsilon \\[4mm]
\dfrac{V(1)}{\min(\text{Deg}_{xy})} \left(r_{ij}^{max} - \left(\dfrac{d_{ij}}{2}\right)^2\right)^{D/2} \geq \dfrac{\eta_{ij}^{RA}}{N} - \epsilon
\end{cases} \quad (34)
$$

i.e.,

$$
r_i + r_j - 2\left(\frac{\eta_{ij}^{RA}/N + \epsilon}{V(1)/\max(\text{Deg}_{xy})}\right)^{1/D} \leq d_{ij} \leq 2\sqrt{r_{ij}^{max} - \left(\frac{\eta_{ij}^{RA}/N - \epsilon}{V(1)/\min(\text{Deg}_{xy})}\right)^{2/D}} \quad (35)
$$

In fact, we can make a more accurate approximation in Eq. (30) if the degree distribution of the graph is known in advance. Similarly, we can provide a proof for AA, with the only difference that $Y_k$ in AA becomes $Y_k = \frac{1}{log(\text{Deg}(k))}$ if $k \in \mathcal{N}(i) \cap \mathcal{N}(j)$, and 0 otherwise.

## C.6    NON-DETERMINISTIC CASE FOR LATENT SPACE MODEL

In this section, we extend our previous analysis based on the infinite values of $\alpha$ ($\alpha \to +\infty$) in Eq. (1) with finite $\alpha$. It could help to generalize our conclusion to more diverse graphs with different underlying mechanisms.

The core idea underlying almost all of our previous results has been the computation of the probability of two nodes $i$ and $j$ having a common neighbor. For the deterministic case, this is simply the area of intersection of two hyperspheres, $A(r_i, r_j, d_{ij})$. However, in the non-deterministic case, there is no such intuitive equivalence. Therefore, our primary goal is to find the representation in the latent space model associated with common neighbor $\eta_{ij}$. The analysis is similar with section C.3 Set $\beta = 0$ and $P(i \sim j | d_{ij}) = 1/\left(1 + e^{\alpha}(d_{ij} - r_{ij}^{max})\right)$ depicts the probability of forming an undirected link between $i$ and $j$ ($i \sim j$), influenced by both feature and structure. From the triangle inequality, we have $d_{ik} < d_{ij} + d_{jk}$, and $d_{ik} > \max\{d_{ij} - d_{jk}, d_{jk} - d_{ij}\}$.

Note $\mathcal{N}(i)$ as the set of neighbors of node $i$ and let $Y_k$ be a random variable depending on the position of point $k$, which is 1 if $k \in \mathcal{N}(i) \cap \mathcal{N}(j)$, and 0 otherwise. Therefore, we have:

$$
\begin{aligned}
E\left[Y_k \mid d_{ij}\right] &= P\left(i \sim k \sim j \mid d_{ij}\right) \\
&= \int P\left(i \sim k \mid d_{ik}\right) P\left(j \sim k \mid d_{jk}\right) P\left(d_{ik}, d_{jk} \mid d_{ij}\right) d\left(d_{ij}\right) \\
&= \int_0^{+\infty} \int_{\max\{d_{ij}-d_{jk}, d_{jk}-d_{ij}\}}^{d_{ij}+d_{jk}} \frac{1}{1+e^{\alpha(d_{ik}-r_{ik}^{max})}} \cdot \frac{1}{1+e^{\alpha(d_{jk}-r_{jk}^{max})}} d\left(d_{ik}\right) d\left(d_{jk}\right) \\
&= \int_0^{d_{ij}} \int_{d_{ij}-d_{jk}}^{d_{ij}+d_{jk}} \frac{1}{1+e^{\alpha(d_{ik}-r_{ik}^{max})}} \cdot \frac{1}{1+e^{\alpha(d_{jk}-r_{jk}^{max})}} d\left(d_{ik}\right) d\left(d_{jk}\right) \quad + \\
&\quad \int_{d_{ij}}^{\infty} \int_{d_{jk}-d_{ij}}^{d_{ij}+d_{jk}} \frac{1}{1+e^{\alpha(d_{ik}-r_{ik}^{max})}} \cdot \frac{1}{1+e^{\alpha(d_{jk}-r_{jk}^{max})}} d\left(d_{ik}\right) d\left(d_{jk}\right) \\
&\leq \int_0^{\infty} \int_0^{+\infty} \frac{1}{1+e^{\alpha(d_{ik}-r_{ik}^{max})}} \cdot \frac{1}{1+e^{\alpha(d_{jk}-r_{jk}^{max})}} d\left(d_{ik}\right) d\left(d_{jk}\right) \quad + \\
&\quad \int_{d_{ij}}^{\infty} \int_0^{+\infty} \frac{1}{1+e^{\alpha(d_{ik}-r_{ik}^{max})}} \cdot \frac{1}{1+e^{\alpha(d_{jk}-r_{jk}^{max})}} d\left(d_{ik}\right) d\left(d_{jk}\right) \\
&= \left( \int_0^{\infty} \frac{1}{1+e^{\alpha(d_{ik}-r_{ik}^{max})}} d\left(d_{ik}\right) + \int_{d_{ij}}^{\infty} \frac{1}{1+e^{\alpha(d_{ik}-r_{ik}^{max})}} d\left(d_{ik}\right) \right) \cdot \int_0^{+\infty} \frac{1}{1+e^{\alpha(d_{jk}-r_{jk}^{max})}} d\left(d_{jk}\right) \\
&= \left( c_1(r_{ik}^{max}, \alpha) + \int_{d_{ij}}^{\infty} \frac{1}{1+e^{\alpha(d_{ik}-r_{ik}^{max})}} d\left(d_{ik}\right) \right) \cdot c_1(r_{jk}^{max}, \alpha) \quad \texttt{(Notation for brevity)} \\
&= \left( c_1(r_{ik}^{max}, \alpha) + \log\left( \frac{1+e^{\alpha \cdot r_{ik}^{max}}}{e^{\alpha(1+2d_{ij})}} \right) \right) \cdot c_1(r_{jk}^{max}, \alpha) \\
&= e_1(r_{ik}^{max}, r_{jk}^{max}, \alpha) \cdot \left( e_2(r_{ik}^{max}, r_{jk}^{max}, \alpha) - (1+2d_{ij}) \right) \quad \texttt{(Notation for brevity)}
\end{aligned}
$$
$$(36)$$

where $e_1(r_{ik}^{max}, r_{jk}^{max}, \alpha), e_2(r_{ik}^{max}, r_{jk}^{max}, \alpha) > 0$ are independent items which do not vary with $d_{ij}$.

For the sake of brevity, we denote $E\left[Y_k \mid d_{ij}\right]$ as $E\left[Y_k\right]$. It can be easy observed that $\sum_{k=0}^{N} Y_k = \eta_{ij}$, i.e., the common neighbour number (CN) of node $i$ and $j$, denoted as $\eta_{ij}$ here. Through empirical Bernstein bounds (Maurer & Pontil, 2009), we have:

$$
P\left[ \left| \sum_k^N Y_k / N - E\left[Y_k\right] \right| \geq \sqrt{\frac{2 \operatorname{var}_N(Y) \log 2/\delta}{N}} + \frac{7 \log 2/\delta}{3(N-1)} \right] \leq 2\delta \tag{37}
$$

where $\operatorname{var}_N(Y) = \frac{\eta_{ij}(1-\eta_{ij}/N)}{N-1}$ is the sample variance of $Y$. Setting $\epsilon = \sqrt{\frac{2 \operatorname{var}_N(Y) \log 2/\delta}{N}} + \frac{7 \log 2/\delta}{3(N-1)}$.

$$
P\left[ \frac{\eta_{ij}}{N} - \epsilon \leq E\left[Y_k\right] \leq \frac{\eta_{ij}}{N} + \epsilon \right] \geq 1 - 2\delta \tag{38}
$$

Combining Eq. (36) and Eq. (38), we can further obtain that for any $\delta > 0$, with probability at least $1 - 2\delta$,

$$
\frac{\eta_{ij}}{N} - \epsilon \leq E\left[Y_k\right] \leq e_1(r_{ik}^{max}, r_{jk}^{max}, \alpha) \cdot \left( e_2(r_{ik}^{max}, r_{jk}^{max}, \alpha) - (1+2d_{ij}) \right) \tag{39}
$$

i.e.,

$$
d_{ij} \leq \frac{\eta_{ij}/N - \epsilon}{2e_1(r_{ik}^{max}, r_{jk}^{max}, \alpha)} + \frac{1}{2}\left( 1 - e_2(r_{ik}^{max}, r_{jk}^{max}, \alpha) \right) \tag{40}
$$

From the above analysis, we can obtain two conclusions in the non-deterministic model (finite $\alpha$) : 1) An analogous proposition 1, which centers on the fact that the upper bound of $d_{ij}$ decreases as $\eta_{ij}$ increases. 2) We have succeeded in finding an upper bound on $E\left[Y_k\right]$ (with $d_{ij}$) in the latent space model, which allows us to similarly obtain all the conclusion of the deterministic model, i.e., the theoretical analyses derived from our model are not constrained by $\alpha$.

## D  DESCRIPTIONS ON MORE DATASETS

To ensure the diversity and completeness of our analysis, we collect datasets from many different domains, e.g., biology (Watts & Strogatz, 1998; Zhang et al., 2018; Von Mering et al., 2002; Oughtred et al., 2019), transport (Batagelj & Mrvar, 2006; Watts & Strogatz, 1998), web (Ackland et al., 2005; Leskovec et al., 2009; Yang & Leskovec, 2012), academic (Newman, 2006; Leskovec et al., 2007; 2005; Yang & Leskovec, 2012) and social networks (Leskovec & Mcauley, 2012; Leskovec et al., 2010; Richardson et al., 2003). With the domain diversity, the graphs in our datasets also cover a large range of types and sizes. Their types include the weighted and the unweighted, the directed and the undirected. The number of nodes in graphs varies from $10^2$ to $10^6$, while the number of edges varies from $10^3$ to $10^7$. Below is a detailed description of the datasets used:

**Selected datasets**  We first present the selected datasets with ignored factors in the existing benchmarking datasets as mentioned in Section 4.2. The statistics of selected domain datasets are listed in table 3. **Power** (Watts & Strogatz, 1998) is an electrical grid of western US. Each node represents a station, and each edge represents a power line between two connected stations. **Amazon-photo** (Shchur et al., 2018) is a portion of the Amazon co-purchase graph (McAuley et al., 2015) in which nodes symbolize products. The presence of an edge between two nodes suggests that these products are often purchased together. The features of each node are derived from product reviews using a bag-of-words encoding. **Reddit** (Zeng et al., 2019) is an undirected graph from the online discussion forum. The nodes represent a post, while the edges indicate posts commented by the same user. The node features the bag-of-word representation of the forum. **Flicker** (Zeng et al., 2019) is an undirected graph originating from NUS-wide. The nodes represent one image uploaded to Flickr, while the edges indicate two images share some common properties (e.g., same geographic location, same gallery, comments by the same user, etc.). The node features the 500-dimensional bag-of-word representation of the images provided by NUS-wide.

Table 3: Selected Datasets Statistics.

|  | Power | Reddit | Amazon Photo | Flicker |
|---|---|---|---|---|
| #Nodes | 4,941 | 232,965 | 7,650 | 334,863 |
| #Edges | 6,594 | 114,615,892 | 238,162 | 899,756 |
| #Feature | NA | 602 | 745 | 500 |
| Mean Degree | 2.67 | 98.38 | 6.21 | 5.69 |
| Split Ratio | 80/10/10 | 80/10/10 | 80/10/10 | 80/10/10 |
| Domains | Transport | Social | Web | Social |

Table 4: Web Domain Datasets Statistics.

|  | PB | Email-Enron | Amazon Photo | Amazon | Google |
|---|---|---|---|---|---|
| #Nodes | 1,222 | 36,692 | 7,650 | 334,863 | 875,713 |
| #Edges | 16,714 | 183,831 | 238,162 | 899,756 | 5,105,039 |
| Mean Degree | 27.36 | 10.02 | 6.21 | 5.69 | 116.49 |
| Split Ratio | 80/10/10 | 80/10/10 | 80/10/10 | 80/10/10 | 80/10/10 |
| Domains | Web | Web | Web | Web | Web |

**Biology Domain**  **C.ele** (Watts & Strogatz, 1998) is an undirected and unweighted neural network of C. elegans. The nodes represent neurons and each edge represents there exists a pathway between two connected neurons. **E.coli** (Zhang et al., 2018) is an undirected, unweighted metabolic network in E. coli. The nodes represent participating metabolites, and the edges indicate that two connected metabolites can interact. **Yeast** (Von Mering et al., 2002) is an undirected and unweighted protein-protein interaction network in yeast. The nodes represent different kinds of proteins and the edges indicate two proteins can interact. The statistics of biology domain datasets are listed in table 5.

**Transport Domain**    **USAir** (Batagelj & Mrvar, 2006) is a weighted, directed network of US Airlines, in which nodes represents airports while weighted edges indicate airline frequency between two airports. **Power** (Watts & Strogatz, 1998) is an electrical grid of western US. Each node represents a station, and each edge represents a power line between two connected stations. The statistics of transport domain datasets are listed in table 6.

**Web Domain**    **PB** (Ackland et al., 2005) is a directed network of US political blogs, where the nodes represent blogs and the directed edges represent links from one blog to another. **Email-Enron** (Leskovec et al., 2009) is a directed, unweighted graph covering around half a million emails. The nodes represent email addresses, and the edges indicate that an address sends an email to another. **Amazon** (Yang & Leskovec, 2012) is an undirected and unweighted graph constructed by Amazon shopping data. The nodes represent products while the edges indicate two connected nodes are used to be bought together. **Google** (Leskovec et al., 2009) is a directed, unweighted graph of website hyperlinks. The nodes represent websites, while the edges indicate that a website has a link to another. The statistics of web domain datasets are listed in table 4.

**Academic Domain**    **NS** (Newman, 2006) is an unweighted and undirected network of collaboration of researchers in network science, where nodes represent scientists and edges represent collaborating relationships. **Ca-AstroPh** (Leskovec et al., 2007) is an undirected, unweighted collaboration network from the e-print arXiv and covers scientific collaborations between authors papers submitted to Astro Physics category. The nodes represent authors and the edges represent the co-authorship. **Ca-CondMat** (Leskovec et al., 2007) is an undirected, unweighted collaboration network from the e-print arXiv and covers scientific collaborations between authors papers submitted to Condense Matter category. The nodes represent authors and the edges represent the co-authorship. **Cit-HepPh** (Leskovec et al., 2005) is a directed and unweighted citation graph from the e-print arXiv. The nodes represent papers and the edges indicate that a paper is cited by another. **DBLP** (Yang & Leskovec, 2012) is an undirected, unweighted collaboration network from the computer science domain. The nodes represent authors and the edges represent the co-authorship. The statistics of academic domain datasets are listed in table 7.

**Social Domain**    **Facebook** (Leskovec & Mcauley, 2012) is a undirected, unweighted social network. The nodes represent users, and the edges represent a friendship relation between any two users. **Vote** (Leskovec et al., 2010) is a directed, unweighted graph of a wiki vote. The nodes represent participants, while the edges indicate the participants vote for one another. **Epinions** (Richardson et al., 2003) is a directed, unweighted who-trust-whom online social network on which users could decide whether to "trust" each other. Each node represents a user, and each edge indicates whether a user trusts another or not. **Slashdot** (Leskovec et al., 2010) is an directed, weighted social network. The nodes represent users, while the edges indicate whether the user thinks another is a friend or foe. The statistics of social domain datasets are listed in table 8. **Reddit** (Zeng et al., 2019) is an undirected graph from the online discussion forum. The nodes represent a post, while the edges indicate posts commented by the same user. The node features the bag-of-word representation of the forum. **Flicker** (Zeng et al., 2019) is an undirected graph originating from NUS-wide. The nodes represent one image uploaded to Flickr, while the edges indicate two images share some common properties (e.g., same geographic location, same gallery, comments by the same user, etc.). The node features the 500-dimensional bag-of-word representation of the images provided by NUS-wide. **Amazon-photo** (Shchur et al., 2018) is a portion of the Amazon co-purchase graph (McAuley et al., 2015) in which nodes symbolize products. The presence of an edge between two nodes suggests that these products are often purchased together. The features of each node are derived from product reviews using a bag-of-words encoding.

## E    ADDITIONAL RESULTS IN MAIN ANALYSIS

**Additional analysis on whether heuristics offer unique perspectives**    In Section 3.1, we conduct an analysis on the complementary effect of different factors. Additional results on Cora and CiteSeer are shown in Figure 7. More results with Hit@10 metric can be found in Figures 8 and 9. Similar observations can be found.

Table 5: Biology Domain Datasets Statistics.

|  | C.ele | E.coli | Yeast |
|---|---|---|---|
| #Nodes | 297 | 1,805 | 2,375 |
| #Edges | 2,148 | 14,660 | 11,693 |
| Mean Degree | 14.46 | 12.55 | 9.85 |
| Split Ratio | 80/10/10 | 80/10/10 | 80/10/10 |
| Domains | Biology | Biology | Biology |

Table 6: Transport Domain Datasets Statistics.

|  | USAir | Power |
|---|---|---|
| #Nodes | 332 | 4,941 |
| #Edges | 2,126 | 6,594 |
| Mean Degree | 12.82 | 2.67 |
| Split Ratio | 80/10/10 | 80/10/10 |
| Domains | Transport | Transport |

**Additional analysis on the overlooked weakness in GNN4LP models**   In Section 3.4, we conduct an analysis that find the overlooked weakness in GNN4LP models. Additional comparison results with GraphSAGE are shown in Figure **??**. Results of the comparison with GCN are shown in Figure 10. Similar observations can be found.

**Data Analysis**   In section 1, we exhibit the data disparity via the distribution of CN scores on different datasets. In this section, we provide more analysis across GSP, LSP, and FP factors in Figure 11. Similar data disparities can be found. Similar observations can be found.

## F   ADDITIONAL ANALYSIS AMONG DATA FACTORS

In Section 3.1, we analyze the significance of link prediction and the complementary effect among them. In this section, we provide further analysis for integrating all these factors for a holistic view. We ultimately investigate whether those data factors could provide a complete view for link prediction with no neglected factors. Typically, we examine whether heuristics derived from different data factors share similar hard negative pairs. The hard negative pairs, the top-ranked negative pairs, pose the main obstacle to link prediction. The negative pair set can be denoted as $\mathcal{E}_i$ where $i$ indicates the $i$-th heuristic algorithm. We conduct experiments to examine the proportion of hard negative edges remaining after including more data factors. The experimental results are shown in Figure 12. The y-axis is the proportion of remaining hard negative edges. It can be calculated as $\frac{|\cap_{i \in S} \mathcal{E}_i|}{|\mathcal{E}_0|}$, where $S$ is the set of heuristic algorithms, $\mathcal{E}_0$ is the hard negative edges of the basis heuristic algorithm. The x-axis indicates heuristic algorithms derived from different data factors. Looking x-axis from left to right, we gradually add new heuristics into the candidate heuristic set, beginning with a single basic algorithm. We can find that when adding each heuristic with a new factor, e.g., adding CN algorithm, the ideal performance increases substantially, indicating the significance of each factor and its complementary effects. Such observations indicate a complete view of those three factors with no factor neglected.

## G   LIMITATION

It is important to acknowledge certain limitations in our work despite our research providing valuable insights on unveiling the underlying factors across different datasets for the link prediction. Notably, there are a series of heuristic algorithms revolving around each factor while we only focus on a few typical ones. To this end, we show experimental results in Section 3.1. It provides a more concrete discussion on the overlapping of heuristic algorithms revolving around the same factors. Nonetheless, there may still remain potential for some ignored heuristic algorithms.

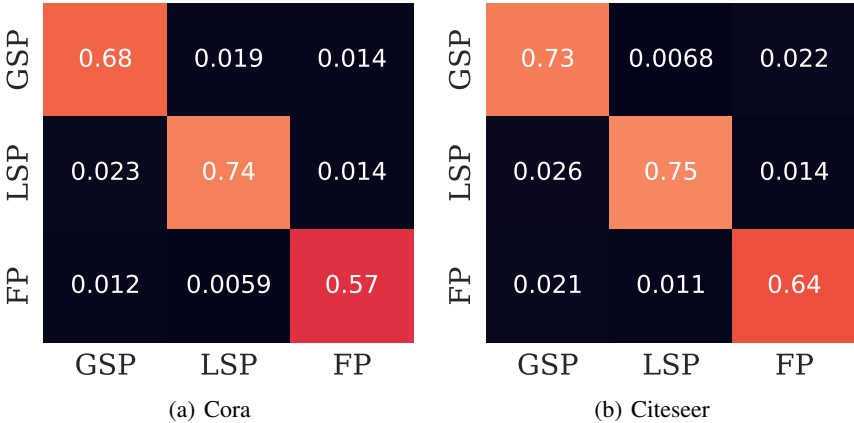

Figure 7: Overlapping ratio between top ranking edges (top 25%) on different heuristic algorithms. Diagonals are the comparison between two heuristics within the same factor, while others compare between heuristics derived from different factors. Experiments are conducted on the hit@10 metric

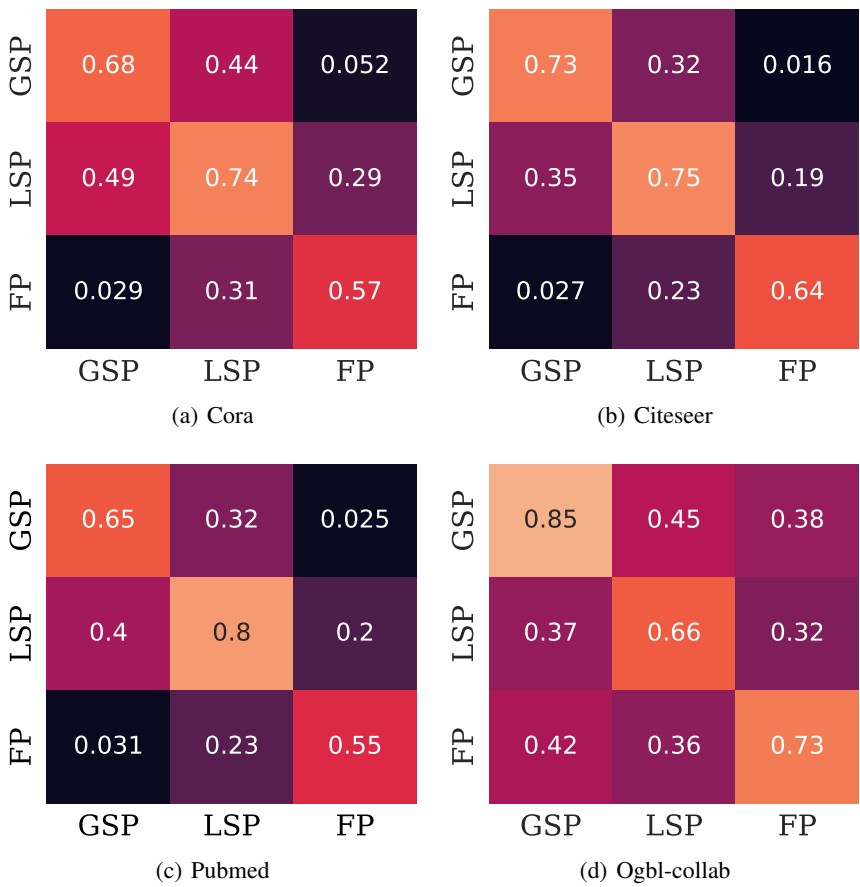

Figure 8: Overlapping ratio between top ranking edges (top 25%) on different heuristic algorithms utilizing Hit@10 metric. Experiments are conducted on Cora, Citeseer, Pubmed and Ogbl-collab. Diagonals are the comparison between two heuristics within the same factor, while others compare between heuristics derived from different factors.

Table 7: Academic Domain Datasets Statistics.

|  | NS | Ca-AstroPh | Ca-ConMat | Cit-HepPh | DBLP |
|---|---|---|---|---|---|
| #Nodes | 1,589 | 18,772 | 23,133 | 34,546 | 317,080 |
| #Edges | 2,742 | 5,105,039 | 93,497 | 421,578 | 1,049,866 |
| Mean Degree | 3.45 | 543.90 | 8.08 | 24.41 | 6.62 |
| Split Ratio | 80/10/10 | 80/10/10 | 80/10/10 | 80/10/10 | 80/10/10 |
| Domains | Academic | Academic | Academic | Academic | Academic |

Table 8: Social Domain Datasets Statistics.

|  | Facebook | Vote | Epinions | Slashdot | Reddit | Flickr |
|---|---|---|---|---|---|---|
| #Nodes | 4,039 | 7,115 | 75,879 | 82,140 | 232,965 | 89,250 |
| #Edges | 88,234 | 103,689 | 508,837 | 549,202 | 114,615,892 | 925,872 |
| Mean Degree | 43.69 | 29.15 | 13.41 | 13.37 | 98.38 | 20.75 |
| Split Ratio | 80/10/10 | 80/10/10 | 80/10/10 | 80/10/10 | 80/10/10 | 80/10/10 |
| Domains | Social | Social | Social | Social | Social | Social |

In order to facilitate a more feasible theoretical analysis, we have made several assumptions, the most significant one is that we separately examine the effect from feature and structure. Typically, we model the feature proximity as an additional noise parameter. The correlation between feature and structure perspectives is only analyzed on the conflict on their effects on link prediction. Such an assumption is subject to a notable drawback since it ignores the correlation between features and existing structural information. It restricts the generality of our theoretical analysis. Moreover, when analyzing the effectiveness of global structural proximity with the number of paths rather than the number of random walks. Paths are deterministic and non-repetitive sequences connecting two points, while random walks are probabilistic sequences with repeat. Extension to the random walks should be the next step.

Our current findings lay a strong foundation for further exploration. It is worth mentioning that our theoretical analysis predominantly focuses on a data perspective. We only show an empirical comparison between vanilla GNN and GNN4LP models, while lacking theoretical analysis from a model perspective. The major obstacle is that the GNN4LP model design is too diverse, making it difficult to analyze comprehensively.

We hope that these efforts will contribute significantly to the Link Prediction research community. Moving forward, we aspire to broaden our findings to encompass more advanced GNN4LP algorithm design, with particular emphasis on the conflict effects on feature similarity and structural similarity. Besides, the guidance we provided on the dataset selection depends entirely on the existing but ignored datasets. We acknowledge that there may still be graphs with distinguishing properties from domains not considered in our work.

## H  BROADER IMPACT

Link prediction is one of the fundamental tasks with multiple applications in the graph domain. Recently, various GNN4LP models have been proposed with state-of-the-art performance. A key aspect of GNN4LP models is to capture pairwise structural information, e.g. neighborhood overlapping features. In this study, we find that, despite GNN4LP models being more expressive to capture pairwise structural patterns, they inherently reveal performance degradation on those edges without sufficient pairwise structural information. Such characteristics may lead to a biased test prediction. For example, if social media utilize GNN4LP models to recommend friends in a social network, introverts with fewer friends will receive low-quality recommendations. Such flaws lead to bad experiences with obstacles for introverts. Therefore, a vicious spiral happens leading to worse situations. Our research offers an understanding of these limitations rather than introducing a new methodology or approach. We identify the overlooked problem and show the potential solution to address this issue. Consequently, we do not foresee any negative broader impacts stemming from

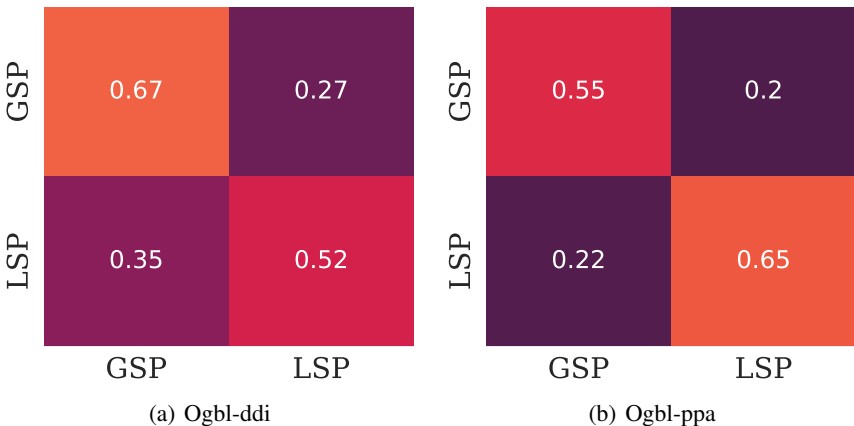

(a) Ogbl-ddi
(b) Ogbl-ppa

Figure 9: Overlapping ratio between top ranking edges (top 25%) on different heuristic algorithms utilizing hit@10 metric. Experiments are conducted on Ogbl-ppa and Ogbl-ddi. Diagonals are the comparison between two heuristics within the same factor, while others compare between heuristics derived from different factors.

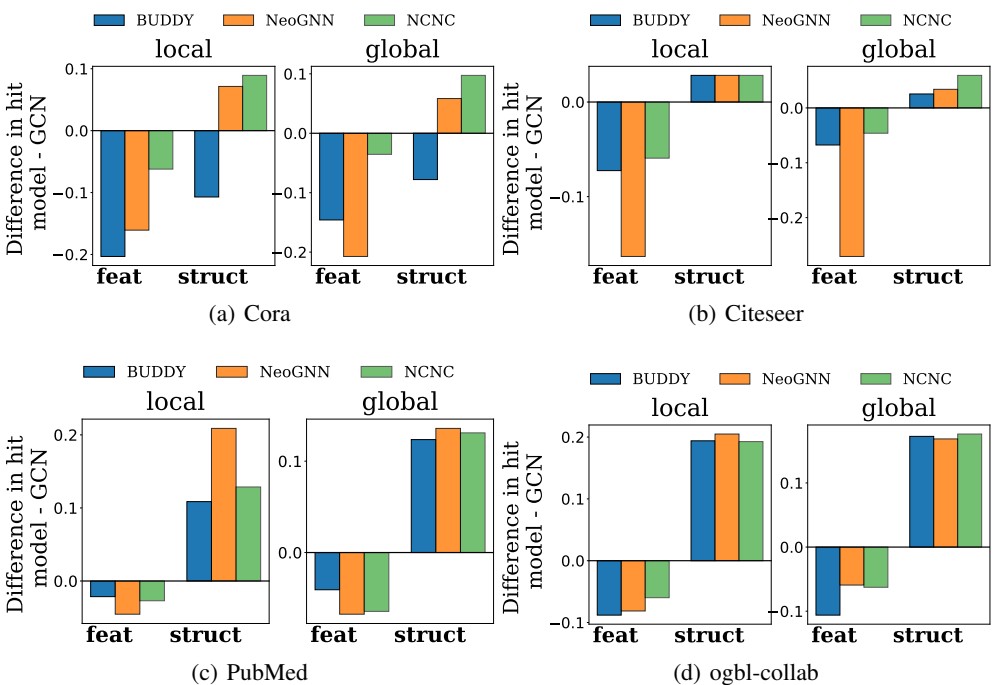

(a) Cora
(b) Citeseer
(c) PubMed
(d) ogbl-collab

Figure 10: Performance comparison between GNN4LP models and vanilla GCN. Two bar groups represent the performance gap on node pairs dominated by feature and structural proximity, respectively. Two sub-figures correspond to compare FP with GSP and LSP, respectively. Vanilla GraphSAGE often outperforms GNN4LP models on node pairs dominated by feature proximity while GNN4LP models outperform on those dominated by structural proximity.

our findings. We expect our work to contribute significantly to the ongoing research efforts aimed at enhancing the versatility and fairness of GNN models when applied to diverse data settings.

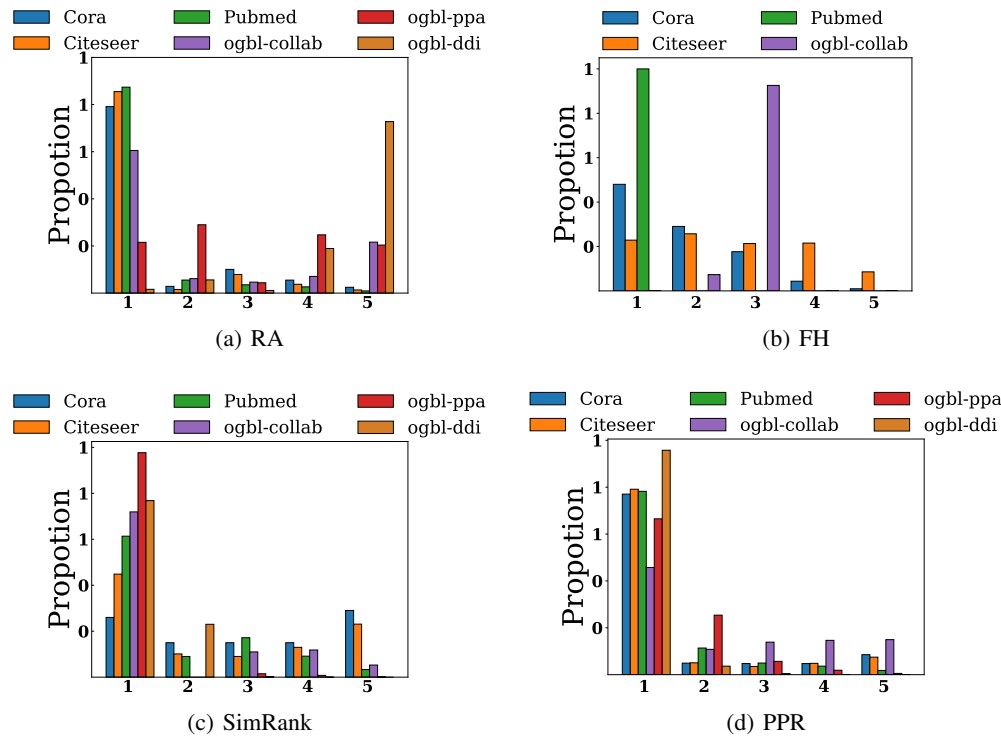

Figure 11: Distribution disparity of different heuristic scores across datasets

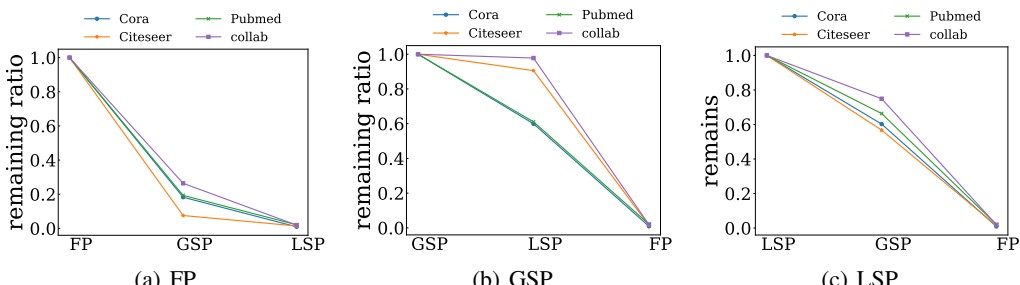

Figure 12: The impact of various heuristic algorithms on the proportion of remaining hard negative edges. The y-axis depicts the remaining hard negative proportion, calculated as $\frac{|\cap_{i\in S}\mathcal{E}_i|}{|\mathcal{E}_0|}$, where $\mathcal{E}_0$ represents hard negative edges from the basic heuristic, the title for each subfigure. Moving from left to right on the x-axis, more heuristics are incrementally added, starting from a basic one.

## I  FUTURE DIRECTION

In section 4.1, we show how our understanding can help guide the model design and dataset selections. It is worth noting that our findings can derive new understandings and inspire us to identify new problems in link prediction with future opportunities. We further illustrate more understanding and initial thoughts on the future of link prediction as follows.

**Dataset categorization for Link Prediction**. In our work, we identify datasets with different data factors based on the performance of the test set, and ease of analysis. However, test edges are not available in real-world scenarios. How to identify data factors becomes an essential problem. One potential direct solution is to utilize the validation set. After identifying the correct direction, we can then select a suitable model for link prediction.

**Specific GNN4LP design for graphs in particular domain**. instead of viewing the link prediction as a whole problem, we suggest decoupling this problem and designing specific models for graphs with different factors. For instance, the specific GNN4LP model should be designed for dense graphs like OGBL-DDI and OGBL-PPA.

**Automatic Machine Learning on Link Prediction**. With specific model designs on graphs with different categories, we can then combine all those components for automatical model architecture selection. The selected model should be able to adaptively decide what type of factors to use, thereby tailoring the pairwise encoding to each individual. It also allows non-experts to benefit from data-driven insights without expert knowledge.

**Inherent connection between node classification and link prediction**. Notably, homophily is a concept popularly discussed in node classification while it is largely ignored in GNN4LP models. The interplay between homophily in link prediction and node classification is a chicken-egg problem. More specifically, link prediction considers that nodes with similar features are likely to be connected. In contrast, node classification considers nodes connected are likely to share similar features. Such a relationship indicates that the principles for node classification and link prediction may benefit from each other. Recently, there has been an attempt (Liu et al., 2023) on utilizing principle in link prediction for node classification. Nonetheless, there is still a lack of exploration on the other side, utilizing the principle of node classification for link prediction.

**Fairness problem in Link Prediction**. In section 3.4, we find the drawback of GNN4LP models over vanilla GNNs. While the overall performance is promising, GNN4LP models may not be satisfying. However, such bias leads to can potentially affect the fairness of the models in real-world applications as discussion in Appendix H. There should be a systematical task design for the fairness problem on link prediction.

**Extension to Network Evolution**. Network evolution (Zaheer & Soda, 2009) is about understanding and modeling how networks change over time. This includes the formation of new links as well as the deletion of existing ones. The link prediction considers the network at a fixed point in time and doesn't account for how the network might evolve and grow temporally. Moreover, link prediction generally only considers a link addition while ignoring the link deletion. We leave extending our understanding of link prediction to network evolution as the future work. The core future steps for extension are:

- link deletion: Despite considering node pairs with high proximity to be more likely to be connected, we need to extend to verify that node pairs with low proximity are more likely to be deleted.

- Temporal feature: Network evolution introduces the time-series feature, considers capturing the time evolution features of link occurrences, and predicts the link occurrence probability. Temporal proximity will be an additional data factor besides the existing three data factors.

**More application in recommendation**. The recommendation (Fu et al., 2022; Wang et al., 2019a; He et al., 2020) can be viewed as a particular link prediction problem on the bipartite graph between users and items. Nonetheless, the graph is generally much more sparse. The mechanism underlying such a specific link prediction problem remains unknown.

## J DATASETS DETAILS

Table 9: Datasets Statistics.

|  | Cora | Citeseer | Pubmed | ogbl-collab | ogbl-ddi | ogbl-ppa |
|---|---|---|---|---|---|---|
| #Nodes | 2,708 | 3,327 | 18,717 | 235,868 | 4,267 | 576,289 |
| #Edges | 5,278 | 4,676 | 44,327 | 1,285,465 | 1,334,889 | 30,326,273 |
| Mean Degree | 3.9 | 2.81 | 4.74 | 10.90 | 625.68 | 105.25 |
| Split Ratio | 85/5/10 | 85/5/10 | 85/5/10 | 92/4/4 | 80/10/10 | 70/20/10 |
| Domains | Academic | Academic | Academic | Academic | Chemistry | Biology |

The basic dataset statistics are shown in Table 9. Notably, three planetoid datasets, CORA, CITESEER, and PUBMED are smaller toy datasets while OGB datasets (Hu et al., 2020) have much more nodes and edges. We include most of the OGB datasets except for ogbl-citation2. The reason is that the standard evaluation method for ogbl-citation2 is different from other datasets (1) Most datasets adopt a shared negative sample setting where all positive samples share the same negative sample set. In contrast, each positive sample in ogbl-citation2 corresponds to an individual negative sample set. (2) The ogbl-citation2 dataset utilizes MRR as the default evaluation metric rather than Hit@K in other datasets. Moreover, we can still achieve a convincing empirical conclusion without ogbl-citation2 since there still remain four citation graphs with varied sizes.

For the data split, we adapt the fixed split with percentages 85/5/10% for planetoid datasets, which can be found at https://github.com/Juanhui28/HeaRT. For OGB datasets, we use the fixed splits provided by the OGB benchmark (Hu et al., 2020).

## K    EXPERIMENTAL SETTINGS & MODEL PERFORMANCE

### K.1    EXPERIMENTAL SETTINGS

We provide implementation details in the following section. Code for our paper can be found in https://anonymous.4open.science/r/LinkPrediction-5EC1/. Notably, we adopt all the settings and implementation from the recent Link Prediction benchmark (Li et al., 2023) for a fair comparison. Codes can be found at https://github.com/Juanhui28/HeaRT. More details are shown as follows.

**Training Settings**. The binary cross entropy loss and Adam optimizer (Kingma & Ba, 2014) are utilized for training. For each training positive sample, we randomly select one negative sample for training. Each model is trained with a maximum of 9999 epochs with the early stop training strategy. We set the early stop epoch to 50 and 20 for planetoid and OGB datasets, respectively.

**Evaluation Settings** For OGB datasets, we utilize the default evaluation metrics provided by (Hu et al., 2020), which are Hits@50, Hits@20, Hits@100 for OGBL-COLLAB, OGBL-DDI, and OGBL-PPA datasets, respectively. For planetoid datasets, we utilize Hit@10 as the evaluation metric.

**Hardware Settings** The experiments are performed on one Linux server (CPU: Intel(R) Xeon(R) CPU E5-2690 v4 @2.60GHz, Operation system: Ubuntu 16.04.6 LTS). For GPU resources, eight NVIDIA Tesla V100 cards are utilized The Python libraries we use to implement our experiments are PyTorch 1.12.1 and PyG 2.1.0.post1.

**Hyperparameter Settings**. For deep models, the hyparameter searching range is shown in Table 10. The general hyperparameter search space is shown in Table 10. The weight decay, number of model and prediction layers, and the embedding dimension are fixed. Notably, several exceptions occur in the general search space resulting in significant performance degradations. Adjustments are made with the guidance of the optimal hyperparameters published in the respective source codes. This includes:

- NCNC (Wang et al., 2023): When training on OGBL-DDI, we adhere to the suggested optimal hyperparameters used in the source code.[2] Specifically, we set the number of model layers to be 1, and we don't apply the pretraining for NCNC to facilitate a fair comparison.

- SEAL (Zhang & Chen, 2018): Due to the computational inefficiency of SEAL, when training on CORA, CITESEER and PUBMED, we further fix the weight decay to 0. Furthermore, we adhere to the published hyperparameters [3] and fix the number of model layers to be 3 and the embedding dimension to be 256.

- BUDDY (Chamberlain et al., 2023): When training on OGBL-PPA, we incorporate the RA and normalized degree as input features while excluding the raw node features. This is based on the optimal hyperparameters published by the authors.[4]

---

[2]https://github.com/GraphPKU/NeuralCommonNeighbor/
[3]https://github.com/facebookresearch/SEAL_OGB/
[4]https://github.com/melifluos/subgraph-sketching/

Table 10: Hyperparameter Search Ranges

| Dataset | Learning Rate | Dropout | Weight Decay | # Model Layers | # Prediction Layers | Embedding Dim |
|---|---|---|---|---|---|---|
| Cora | (0.01, 0.001) | (0.1, 0.3, 0.5) | (1e-4, 1e-7, 0) | (1, 2, 3) | (1, 2, 3) | (128, 256) |
| Citeseer | (0.01, 0.001) | (0.1, 0.3, 0.5) | (1e-4, 1e-7, 0) | (1, 2, 3) | (1, 2, 3) | (128, 256) |
| Pubmed | (0.01, 0.001) | (0.1, 0.3, 0.5) | (1e-4, 1e-7, 0) | (1, 2, 3) | (1, 2, 3) | (128, 256) |
| ogbl-collab | (0.01, 0.001) | (0, 0.3, 0.5) | 0 | 3 | 3 | 256 |
| ogbl-ddi | (0.01, 0.001) | (0, 0.3, 0.5) | 0 | 3 | 3 | 256 |
| ogbl-ppa | (0.01, 0.001) | (0, 0.3, 0.5) | 0 | 3 | 3 | 256 |

## K.2 MODEL PERFORMANCE

We include baseline models that are both representative and scalable to most datasets. Models like NBFNet (Zhu et al., 2021b) are not included since they meet severe out-of-memory issue on those larger OGB datasets. The detailed model performance is shown in Table 11. Since the ogbl-ddi has no node features, we mark the MLP results with a "N/A". Notably, there is no consistent winning solution across different settings. All results except for SimRank, PPR, and FH are from Li et al. (2023).

Table 11: Model performance on Cora, CiteSeer, Pubmed, and OGB datasets.

| | Cora | Citeseer | Pubmed | Ogbl-collab | Ogbl-ddi | Ogbl-ppa |
|---|---|---|---|---|---|---|
| CN | 42.69 | 35.16 | 27.93 | 61.37 | 17.73 | 27.65 |
| AA | 42.69 | 35.16 | 27.93 | 64.17 | 18.61 | 32.45 |
| RA | 42.69 | 35.16 | 27.93 | 63.81 | 6.23 | 49.33 |
| Katz | 51.61 | 57.36 | 42.17 | 64.33 | 17.73 | 27.65 |
| SimRank | 50.28 | 52.52 | 37.03 | 21.75 | OOM | OOM |
| PPR | 32.96 | 31.88 | 32.84 | 54.53 | 7.19 | 4.97 |
| FH | 47.13 | 48.19 | 34.57 | 23.92 | 4.74 | NA |
| MLP | $53.59 \pm 3.57$ | $69.74 \pm 2.19$ | $34.01 \pm 4.94$ | $35.81 \pm 1.08$ | N/A | $0.45 \pm 0.04$ |
| GCN | $66.11 \pm 4.03$ | $74.15 \pm 1.70$ | $56.06 \pm 4.83$ | $54.96 \pm 3.18$ | $49.90 \pm 7.23$ | $29.57 \pm 2.90$ |
| SAGE | $63.66 \pm 4.98$ | $78.06 \pm 2.26$ | $48.18 \pm 4.60$ | $59.44 \pm 1.37$ | $49.84 \pm 15.56$ | $41.02 \pm 1.94$ |
| NeoGNN | $64.10 \pm 4.31$ | $69.25 \pm 1.90$ | $56.25 \pm 3.42$ | $66.13 \pm 0.61$ | $20.95 \pm 6.03$ | $48.45 \pm 1.01$ |
| BUDDY | $59.47 \pm 5.49$ | $80.04 \pm 2.27$ | $46.62 \pm 4.58$ | $64.59 \pm 0.46$ | $29.60 \pm 4.75$ | $47.33 \pm 1.96$ |
| NCNC | $75.07 \pm 1.95$ | $82.64 \pm 1.40$ | $61.89 \pm 3.54$ | $65.97 \pm 1.03$ | $70.23 \pm 12.11$ | $62.64 \pm 0.79$ |

