# OpenReview forum: "Revisiting Link Prediction: a data perspective"
_ICLR.cc/2024/Conference — ICLR 2024 poster_

### Official Review · Reviewer_cwp2 · 2023-10-21

**Soundness:** 3 good
**Presentation:** 1 poor
**Contribution:** 3 good
**Rating:** 6
**Confidence:** 4

**Summary:**

The paper analyzes link prediction across diverse datasets from a data-centric perspective. Specifically, it considers that a link can be generated through Local structural proximity (LSP), Global structural proximity (GSP), and Feature proximity (FP). The experiments show discrepancies in the top-ranked edges according to the selected features. The paper suddenly changes to the proposition of a model based on feature proximity, latent node pairwise distance, and some hyperparameters. The paper supports this model claiming that node pairs with a large number of common neighbors tend to have low feature proximity and vice versa, and that this phenomenon is given once the network has been already formed over time. Finally, the paper suggests considering the feature space and structural space by different models and combining them at the end.

**Strengths:**

The paper shows a very significant result and contribution, that LSP, GSP, and FP should be considered differently when they are used for link prediction. More details about this conclusion are given, and it will be important to clearly define the time in this conclusion. As it is mentioned in the paper, the claim is based on a given network, that has already made several connections among them.

The proposition of the new theoretical model is interesting but not so novel. I mean, while the combination of LSP, GSP, and FP has been largely considered, the novelty is the inclusion of these characteristics in a different factor (more important to FP when nodes have a large distance and vice versa).

**Weaknesses:**

Readability. As it is stated in the submission instructions "reviewers are not required to read the appendix". In this case, I did not read the appendix. For this reason, I could understand specific details from the paper. While I understand the main idea, there are details, such as the parameter r that I could not understand. Unfortunately, the paper uses the appendix to explain very important parts of the paper; for example: "Limitation, broader impact, and future works are in Appendix G, H, and I". So, if you can not reduce and contain the paper to 9 pages, I suggest you submit the paper to a journal, instead of a conference. Note, I usually read the appendices of the paper. They usually have important proof and complementary results, but in this case, it is being used as a main part of the paper.

The details of the theoretical models are not explained in the main paper. There are several details without any type of information, for example, the parameter r. It seems then rather than a parameter, r is a hyperparameter, but no details about this are given in the main paper. I have similar issues with the \beta parameter/hyperparameter, which is described as a feature proximity parameter, but no details about its calculation or suggestion are given.

I understand the main idea of the theoretical model and its importance in the paper, but this is not used in the experiment section of the main paper.

**Questions:**

Can you make the paper readable by itself using 9 pages in length?
Thanks for changing the paper.

---

> ### Author Response · Authors · 2023-11-14
> **Response to reviewer cwp2(1/3)**
>
> **Response:**
> We thank the reviewer for the concerns on the readability. We are also glad to see that the reviewer appreciates "our paper showing a very significant result and contribution".
>
> For the readability issue, we worked hard to improve the writing to make the revision self-contained. **The reviewer can directly see our new updated revision that is better self-contained. If you still have any concerns, please let us know immediately, and we will make sure that the paper is self-contained with clear writing.** In the following part, we first summarize your questions, explain them in detail, and show our improvement in the revision. Besides, we carefully check all the appendix sections to avoid missing any important parts of the paper in the main content. If you still have any concerns about the readability, please let us know. We are eager to **make sure the paper is self-contained within the 9-page limitation to meet the standard for ICLR.**
>
> **Questions are summarized and retold as follows**: Questions are majorly three-fold: (1) The main paper lacks some important details about parameters $r$ and $\beta$. (2) Limitation, broader impact, and future works are in Appendix G, H, and I" (3) I understand the main idea of the theoretical model and its importance in the paper, but this is not used in the experiment section of the main paper. We then provide explanations and the corresponding revision to those three questions step by step.
>
> **Question1:** The main paper lacks some important details about the parameter $r$ and $\beta$.
>
> **Response1:** Thanks for pointing out this issue. In the original paper, we put most detailed explanations of symbols and parameters in Section 3.2 when we first introduce the latent space model. However, we do not repeat the detailed meanings of those parameters or notify the reader where they can find the detailed explanations in each lemma and proposition. We have made the change in the modification accordingly to avoid potential confusion.
>
> **The detailed explanation of those parameters is as follows.**  We first clarify that all those parameters are only utilized for the theoretical analysis, corresponding to different intrinsic properties of datasets. They show no relevance in training a deep model process. Therefore, all those stuff in the theoretical analysis should be called parameters, not hyperparameters.
>
> **Detail explanation for parameter $r$ is as follows.**
> > $r_i$ is a connecting threshold parameter corresponding to node $i$.With $\alpha = +\infty$, $\frac {1}{1+ e^ {\alpha (d_{ij}-\max\{r_{i},r_{j}\})}}=0$ if $d_{ij} > \max\{r_i, r_j\}$, otherwise it equals to $1$. Therefore, a large $r_i$ indicates node $i$ is more likely to form edges, leading to a potentially larger degree. Nodes in the graph can be associated with different $r$ values, allowing us to model graphs with various degree distributions. Such flexibility enables our theoretical model to be applicable to more real-world graphs.
>
> The detailed discussion of parameter $r$ can be found at the beginning of Section 3.2 theoretical model.
>
> **Detail explanation for parameter $\beta$ is as follows.**
> > Feature perspective provides complementary information, considering two nodes with high feature proximity but located distantly in the latent space should also be potentially connected. In line with this, we introduce the feature proximity parameter $\beta_{ij}$ in the latent space. A larger $\beta_{ij}$ indicates more likely for node $i$ and $j$ to be connected.
>
> The detailed discussion of parameter $\beta$ in Section 3.2 can be found on, the 4th line of page 5, starting with **(ii)**. Moreover, $\beta$ is a feature proximity parameter in the theoretical model which does not have a direct calculation formulation. It is just for the theoretical analysis. In the experiment part, we calculate the heuristic method, FH, corresponding to feature proximity $\beta$ for calculation. Details in the paper are shown in Section 3.1.
>
> **Our modification in the revision:** We carefully check whether there is a description each time we mention each parameter. We add a simple description **each time** we mention the parameter and let the reader see more details in the corresponding part in section 3.2 of the main paper. The newly added content is as follows
> > $r_i$ is a connecting threshold parameter corresponding to node $i$ while a large $r_i$ indicates node $i$ is more likely to form edges. A more detailed description of $r$ can be found under Eq.(1). $\beta_{ij}$ is the feature proximity parameter between nodes $i$ and $j$ in the latent space while a large $\beta_{ij}$ indicates large feature proximity. A more detailed description of $\beta$ can be found in Section 3.2.

---

> > ### Author Response · Authors · 2023-11-14
> > **Response to reviewer cwp2(2/3)**
> >
> > **Question2:** "Limitation, broader impact, and future works are in Appendix G, H, and I"
> >
> > **Response2:** We understand and agree with your concern about the central role of appendices in our manuscript. We further agree with you that the appendices mainly contain supplementary analyses, extended discussions, and additional data that support the main findings without being central to the paper. In our paper, we discuss the limitations in our paper and broader impact honestly and comprehensively. These parts also provide insight and make our contribution more solid. **In the new revision, we moved those key concepts to the main paper, making the paper self-contained.**
> >
> >
> > **Our modification in the revision:** We carefully added the key concepts of limitation, broader impact, and future works in the main content. You can find it in the conclusion section. Below are the new content we added to the paper.
> >
> > > In this work, we explore link prediction from a data perspective, elucidating three pivotal factors: LSP, GSP, and FP. Theoretical analyses uncover the underlying incompatibility. Inspired by such incompatibility, our paper shows a positive broader impact as we identify the overlooked biased prediction in GNN4LP models and show the potential solution to address this issue. Our understanding provides guidance for the new model design and how to select benchmark datasets for comprehensive evaluation. Such understanding also gains insights for future direction including (1) adding a more careful discussion on the above fairness issue and (2) designing specific GNN4LP models for datasets in different dataset categories mentioned in Sec 4.2. Nonetheless, our paper shows minor limitations as we make the assumption that the feature proximity is an additional noise parameter rather than adaptively combining that information in the same subspace in theoretical analysis. A more comprehensive discussion on Limitation, broader impact, and future works are in Appendix G, H, and I.

---

> > > ### Author Response · Authors · 2023-11-14
> > > **Response to reviewer cwp2(3/3)**
> > >
> > > **Question3:** I understand the main idea of the theoretical model and its importance in the paper, but this is not used in the experiment section of the main paper.
> > >
> > > **Response3:** Thanks for your question on the connection between the theoretical model and the experiment in the paper. Our primary answers are (1) for all the theories, we have the corresponding experimental results to support our claims. (2) **We indeed utilize the qualitative insight we gain from the theoretical analysis to guide the model design and dataset selection.** We have made changes in the revision accordingly to make it clear and easy to understand.
> > >
> > > We first show our theoretical insights and how they relate to experiments. Our theoretical insights are majorly two-fold: (1) Global structural proximity (GSP) only shows effectiveness when local structural proximity (LSP) is deficient. (2) The incompatibility between LSP and feature proximity (FP). It is unlikely that both large LSP and FP co-exist for a single node pair. Those insights guide the experiment design in section 3.4 and 4.
> > >
> > > In Section 3.4, we study how the incompatibility between structural proximity and feature proximity affects the effectiveness of GNN4LP models. In this section, we are majorly inspired by insight (2): the incompatibility between structural and feature factors. Due to such incompatibility, it could be hard for a single model to benefit both node pairs with feature proximity factors with opposite information. Therefore, we observe that the GNN4LP with specific designed for structural proximity will perform better on the node pairs with structural proximity but not the feature ones
> > >
> > > In Section 4.1, we study the guidance for the model design inspired by insight (2): the incompatibility between structural and feature factors. Since we observe when both structural and feature factors come into play simultaneously, there is a potential for them to provide conflicting supervision to the model. Then we propose the decouple the model design which learns the feature proximity factors and pairwise structural ones independently before integrating their outputs, to avoid the conflict from insight 3.
> > >
> > > In Section 4.2, we illuminate the complete dataset landscape inspired by insight (1): The LSP and GSP can not be effective simultaneously in the same dataset. To get the full landscape, we enumerate all potential combinations among different data factors. As there are three factors, the potential combination should be $C_3^3 + C_3^2 + C_3^1=7$. Nonetheless, given the constraint from insight (1), the potential combinations shrink with only four combinations: Category 1: both LSP and FP factors dominate. Category 2: Only the LSP factor dominates. Category 3: both GSP and FP factors dominate. Category 4: Only the GSP factor dominates. The theoretical insight helps us to clearly identify the different types of datasets in the link prediction properly, avoiding those inexistent categories.
> > >
> > > We believe that the **qualitative insights** from theoretical analysis indeed provide insights for the experiment part, especially for the model design and dataset selection.
> > >
> > >
> > > **Our modification in the revision:** We carefully added the key concepts of the writing when we connected the theoretical limitation, broader impact, and future works in the main content. **We add more intuition about the connection between the theory insight and experiment at the beginning of each subsection. Please see the revision.**
> > >
> > >
> > > We are committed to further enhancing the accessibility and clarity of our manuscript and appreciate your constructive feedback in this regard. Thank you for your consideration and guidance. **Moreover, we carefully check all the appendix sections to avoid missing any important parts of the paper in the main content. Hope that the reviewer finds the revision meets the requirement to be self-contained in 9 pages. If you have any further questions, feel free to mention them to us, and we will make modifications immediately. **

---

> > > > ### Author Response · Authors · 2023-11-17
> > > > **A kind reminder**
> > > >
> > > > Dear reviewer cwp2,
> > > >
> > > > Thank you for taking the time to review our work. We appreciate your feedback and we have prepared a thorough response to address your concerns. We believe that we have responded to and addressed all your concerns with our revisions — in light of this, we hope you consider raising your score. Please let us know in case there are outstanding concerns, and if so, we will be happy to respond.
> > > >
> > > > Notably, given that we are approaching the deadline for the rebuttal phase, we hope we can have the discussion soon. Thanks.

---

> ### Author Response · Authors · 2023-11-22
> **The last day remind for reviewer cwp2**
>
> Dear reviewer cwp2, as today is the final day of our discussion, we appreciate the opportunity to engage with you. If you have any remaining questions or concerns, please don't hesitate to share them with us, we will be happy to respond. Thank you.

---

### Official Review · Reviewer_wtEd · 2023-10-28

**Soundness:** 3 good
**Presentation:** 3 good
**Contribution:** 2 fair
**Rating:** 6
**Confidence:** 5

**Summary:**

This is a benchmark paper on link prediction. The authors experiment comprehensively and come to some interesting obervations. In the first part, the authors analyze the data perspective from three heuristics (1)Local structural proximity(LSP) (2)Global structural proximity(GSP) (3) Feature proximity(FP). Two natural questions are studied: how much does each help and whether they have overlappings. Then the authors introduce a latent space model and then theoretically demonstrate that the model reflects the effectiveness of LSP, GSP, and FP factors.  Then, based on the undetstandings of the heuristic groups, the authors provide practical guidelines from model and data perspective.

**Strengths:**

1. The writing is easy to follow and the paper presentation is clear
2. The overall motivation is clear and the work is comprehensive. Both experiments and analysis look good.

**Weaknesses:**

I'd thank the authors to elaborate more on my questions below.

**Questions:**

1. What is the role of heterophily when calculating Feature proximity? Moreover, could you give an example in homophilic graph that FP and LSP are conflictive or FP and GSP are conflictive. I am asking this because I feel the conflict comes from the different link forming mechanism where FP is just not appropriate.
2. I am not quite sure but I think NeurIPS D&B track and TMLR might be more appropriate venues.

---

> ### Author Response · Authors · 2023-11-14
> **Response to reviewer wtEd**
>
> **Question1:** What is the role of heterophily when calculating Feature proximity? Moreover, could you give an example in the homophilic graph that FP and LSP are conflictive or FP and GSP are conflictive. I am asking this because I feel the conflict comes from the different link-forming mechanisms where FP is just not appropriate
>
>
> **Response:** Thanks for your question about the role of heterophily when calculating feature proximity. We are not quite sure whether we exactly understand some details of your question. Therefore, we will first retell your question and answer it accordingly. If there exists any misunderstanding, please let us know, we are happy to solve all your valuable concerns and improve our paper accordingly.
>
> Our current understanding of your question is two-fold: (1) How to describe the heterophily with the feature proximity factor? and (2) Does the conflict only happen when low FP(heterophily) but high LSP? How can the conflict happen when high FP(homophily) but low LSP? Can you show an example?
>
> **R1:** Heterophily is the opposite to the well-known homophily concept in social science [1] with the principle "similarity breeds connection". More specifically, homophily indicates that nodes with larger feature similarity (high feature proximity). In contrast, heterophily indicates smaller feature similarity (small feature proximity). The ogbl-ddi dataset is the typical example where most node pairs are feature heterophilic as indicated in Figure 2.
>
> **R2:** No, the conflict can also happen in the homophily case with high FP(homophily) and low LSP.
> - We first clarify that there exists the case where node pairs with high FH but low LSP. The theoretical evidence and the empirical verification can be found in Lemma 2 and Figure 4, respectively. In Figure 4, we can observe that there is a large proportion of test node pairs that can be predicted correctly by heuristic-derived feature proximity (high FP) but predicted wrongly by heuristic derived local structural proximity (low LSP).
> - We then explain more on why there exists the case feature-homophilic pairs that FP and LSP are conflictive. **The key concept is, that link prediction is to predict the node pairs that aren't connected yet, excluding those node pairs that are already connected. When node pairs have both high FP and LSP, they are more likely to be with a link that exists already, rather than still unconnected for prediction.** We are then to predict those unconnected node pairs without both high FP and LSP. In such cases, high FP or high LSP individually is strong evidence of a potential link connected. Therefore, in a homophilic graph, it is likely that the unconnected node pairs with high FP and low LSP(conflictive) have a high probability of being connected.
> - We want to clarify that in the link prediction task, feature proximity and structural proximity are all important, rather than the feature proximity dominating this problem. Evidence can be found in proposition 1 and 3. The feature and structure are two perspectives that may not necessarily be aligned with each other. Similar thoughts can also be found in social network analysis[2].
>
>
> [1] McPherson, M., Smith-Lovin, L., & Cook, J. M. (2001). Birds of a feather: Homophily in social networks. Annual Review of Sociology.
> [2] Abebe, Rediet, et al. "On the effect of triadic closure on network segregation." Proceedings of the 23rd ACM Conference on Economics and Computation. 2022.
>
>
> **Weakness2:** I am not quite sure but I think NeurIPS D&B track and TMLR might be more appropriate venues.
>
> **Response:** Thank you for suggesting alternative venues for our paper, including NeurIPS D&B track and TMLR. We appreciate and consider your suggestion according to our paper's focus and contributions. We agree that the NeurIPS D&B track and TMLR are prestigious venues with a strong focus on deep learning and machine learning research. **We still believe that our paper is also well-suited for ICLR.** The reason is that, despite our analysis being from a data perspective, we are not limited to empirical findings revolving on the new suitable datasets and benchmarking existing algorithms. Our paper focuses more on describing data with theoretical analyses derived from deep insight into the link prediction task.** Such understandings align with both the model perspective** (models to capture diverse pairwise data patterns, e.g., local structural patterns, the number of paths, and structural position) **and the data perspective** (data dominated by three factors: FP, LSP, and GSP). Our paper brings innovative understandings in the field of link prediction, a topic that aligns well with ICLR's focus on cutting-edge research in deep learning.

---

> > ### Author Response · Authors · 2023-11-17
> > **A kind reminder**
> >
> > Dear reviewer wtEd,
> >
> > Thank you for taking the time to review our work. We appreciate your feedback and we have prepared a thorough response to address your concerns. We believe that we have responded to and addressed all your concerns with our revisions — in light of this, we hope you consider raising your score. Please let us know in case there are outstanding concerns, and if so, we will be happy to respond.
> >
> > Notably, given that we are approaching the deadline for the rebuttal phase, we hope we can have the discussion soon. Thanks!

---

> ### Author Response · Authors · 2023-11-22
> **The last day remind for reviewer wtEd**
>
> Dear reviewer wtEd, as today is the final day of our discussion, we appreciate the opportunity to engage with you. If you have any remaining questions or concerns, please don't hesitate to share them with us, we will be happy to respond. Thank you.

---

### Official Review · Reviewer_Rht2 · 2023-10-29

**Soundness:** 4 excellent
**Presentation:** 3 good
**Contribution:** 4 excellent
**Rating:** 8
**Confidence:** 5

**Summary:**

This paper explores the principles of link prediction in graphs from a data-centric perspective. It identifies three critical factors for link prediction: local structural proximity, global structural proximity, and feature proximity. The paper reveals relationships among these factors and highlights an incompatibility between feature proximity and local structural proximity. This insight has implications for the performance of Graph Neural Networks for Link Prediction (GNN4LP) models. The study offers practical guidance for model design and benchmark dataset selection in the field of link prediction.

**Strengths:**

1.  The paper takes a unique data-centric approach to the problem of link prediction, shifting the focus from model design to understanding the underlying data factors. This perspective is essential for providing valuable insights into the link prediction task and guiding future research.
2.  The paper identifies and empirically validates three critical factors for link prediction: local structural proximity, global structural proximity, and feature proximity. This systematic analysis enhances our understanding of what influences link formation in graphs, contributing to a more comprehensive approach to link prediction.
3. The revelation of an incompatibility between feature proximity and local structural proximity in link prediction is a significant finding. It highlights a vulnerability in existing GNN4LP models and suggests that certain links, primarily driven by feature similarity, may be challenging to predict accurately. This insight provides researchers with a new perspective on model limitations and opportunities for improvement.

**Weaknesses:**

1. The paper introduces a latent space model for link prediction and provides theoretical guarantees for the identified data factors. Nonetheless, why to use such model is still so clear and motivated
2. Instead of hit metric, mrr is also very important for link prediction. It could be better if there is additional analysis on MRR
3. The work somehow shows correlation with the network evolution. Can you add some detailed discussion with the network evolution especially the dynamic case?

**Questions:**

see the weaknesses

---

> ### Author Response · Authors · 2023-11-14
> **Response to reviewer Rht2(1/2)**
>
> **Weakness1:** The paper introduces a latent space model for link prediction and provides theoretical guarantees for the identified data factors. Nonetheless, why using such a model is still so clear and motivated?
>
> **Response:** Thanks for your great question about the motivation for the latent space model. The primary answer is that the latent space model is particularly suited for link prediction as it connects the likelihood of a link forming between two nodes to their proximity in a latent metric space. It is developed based on the most central principle in link prediction: Similarity breeds connection [6]. That is, there is a higher probability of forming a link if two nodes have similar characteristics.
>
> The latent space model is a widely adopted **theoretical model** that helps design many real-world problems, especially for network analysis [1,2,3,4] and physics. Particularly, the latent space model in network analysis aims to locate network information in some latent space that can describe both local and global structures. The selected latent space can be Euclidean space [1], unit hypersphere space [5], and so on to describe different properties.
>
> We then utilize the latent space model for link prediction, based on the important data factors, considering both feature and structure perspectives comprehensively. Each node is associated with a location in a D-dimensional latent space. The latent space for link prediction is typically utilized to describe node proximity mathematically, where nodes close in the latent space are likely to share particular characteristics. We can easily connect the distance in the latent space with the probability of forming a link. And the distance in the latent space can be approximated with different data factors. The theoretical model is utilized to mathematically formulate the graph data and link prediction task. Such mathematical description can help us conduct theoretical analysis and bring insights into the effectiveness of these data factors (Proposition 1,2,3) and the relationship between different factors (Lemma 1,2).
>
> [1] Peter D Hoff, Adrian E Raftery, and Mark S Handcock. Latent space approaches to social network analysis. Journal of the american Statistical association, 97(460):1090–1098, 2002.
> [2] Sewell, Daniel K., and Yuguo Chen. "Latent space models for dynamic networks." Journal of the american statistical association 110.512 (2015): 1646-1657.
> [3] Smith, Anna L., Dena M. Asta, and Catherine A. Calder. "The geometry of continuous latent space models for network data." Statistical science: a review journal of the Institute of Mathematical Statistics 34.3 (2019): 428.
> [4] Gormley, Isobel Claire, and Thomas Brendan Murphy. "A latent space model for rank data." ICML Workshop on Statistical Network Analysis. Berlin, Heidelberg: Springer Berlin Heidelberg, 2006.
> [5] Jiang, Diqiong, et al. "Sphere Face Model: A 3D morphable model with hypersphere manifold latent space using joint 2D/3D training." Computational Visual Media 9.2 (2023): 279-296.
> [6] McPherson, M., Smith-Lovin, L., & Cook, J. M. (2001). Birds of a feather: Homophily in social networks. Annual Review of Sociology.
>
> **Weakness2:** Instead of hit metric, mrr is also very important for link prediction. It could be better if there is additional analysis on MRR
>
> **Response**: We agree that MRR is also a very important metric. As such, we also add results with MRR. They are added to the revision in Appendix E: "ADDITIONAL RESULTS IN MAIN ANALYSIS". We observe that our existing conclusions are unchanged via the inclusion of MRR and in fact, are further verified.

---

> > ### Author Response · Authors · 2023-11-14
> > **Response to reviewer Rht2(2/2)**
> >
> > **Weakness3:** The work somehow shows a correlation with the network evolution. Can you add some detailed discussion on the network evolution, especially the dynamic case?
> >
> > **Response:** Thanks for your great question for the discussion about the insights in network evolution and link prediction. We provide the definition of network evolution and link prediction and make a comparison between them. We take the most well-adopted definition as follows. Link prediction is concerned with determining the likelihood of a link forming between two nodes in a network, given its current state. Network evolution is about understanding and modeling how networks change over time. This includes the formation of new links as well as the deletion of existing ones. Link prediction considers the network at a fixed point in time and doesn't account for how the network might evolve and grow temporally. Moreover, link prediction generally only considers a link addition while ignoring the link deletion.
> >
> >
> > Our paper focuses on a static link prediction setting, it is different from the network evolution considering temporal information. We leave extending our understanding of link prediction to network evolution as the future work. The core future steps for extension are:
> > - **link deletion:** The current latent space model considers node pairs with high proximity to be more likely to be connected. Future work can extend the model verifying that node pairs with low proximity are more likely to be deleted.
> > - **Temporal feature:** Network evolution introduces a time component, that further how time impacts the probability of a link occurring. Further work can consider temporal proximity as an additional data factor besides the existing three data factors we examined.
> >
> > The above discussion on the future is added in Appendix I: Future Work.

---

> > > ### Author Response · Authors · 2023-11-17
> > > **A kind reminder**
> > >
> > > Dear reviewer Rht2,
> > >
> > > Thank you for taking the time to review our work. We appreciate your feedback and we have prepared a thorough response to address your concerns. We believe that we have responded to and addressed all your concerns with our revisions — in light of this, we hope you consider raising your score. Please let us know in case there are outstanding concerns, and if so, we will be happy to respond.
> > >
> > > Notably, given that we are approaching the deadline for the rebuttal phase, we hope we can have the discussion soon. Thanks!

---

> > > > ### Comment · Reviewer_Rht2 · 2023-11-20
> > > > **Thanks for the response**
> > > >
> > > > After careful reconsideration and review of the feedback provided by other reviewers, I decide to raise the score accordingly and further champion for this paper. I believe this paper can inspire many future directions in the link prediction and network evolution domains.

---

> > > > > ### Author Response · Authors · 2023-11-21
> > > > > **Thanks**
> > > > >
> > > > > We are glad to know that our rebuttal has addressed your concerns. Thank you very much for your time, and for updating your score.

---

### Official Review · Reviewer_zuoh · 2023-11-06

**Soundness:** 3 good
**Presentation:** 3 good
**Contribution:** 1 poor
**Rating:** 5
**Confidence:** 3

**Summary:**

This paper studies the reasons why GNNs largely fail to deliver on the task conventionally called "link prediction", which is the task of recovering, rather than "predicting", missing links in a graph, from a data-oriented perspective. It suggests than the task of link recovery depends on three factors, namely the local structral proxmimity of nodes, their global structural proximity, and their feature proximity, and examines how these three factors interrelate. The core finding of the study is that conventional GNNs for link prediction fail when the feature proximity factor becomes dominant. Given these insights, the paper offers advice on how one could select data for selecting benchmark data to evaluation GNNs for link recovery.

**Strengths:**

Drawing from previous work, the paper suggests and reconfirms the importance of three key factors underlying the task of link recovery and proposes a latent space model that effectively embeds nodes in a D-dimensional space, which is used to analyze the relationships among the proposed data factors, revealing an underlying tension between feature proximity and local structural proximity, in other words, the fact that co-occurrence of both high feature similarity and high local structural similarity rarely happens. This is an interesting empirical finding.

**Weaknesses:**

The core finding of the paper is the features should also be taken in consideration. This finding has been known to the community, especially when it comes to using embeddings for link recovery. The proposed latent space model looks suspiciously similar to a node embedding used for link recover. As such, it calls for comparison with state-of-the-art works on embeddings for that purpose, which examine pretty much the same issues as this paper. Such a comparison or even discussion of the relationship and underlying novelty is missing.

**Questions:**

Is the proposed latent space different from a node embedding, and if so, why?

---

> ### Author Response · Authors · 2023-11-14
> **Response to reviewer zuoh (1/2)**
>
> **Weakness1:** The core finding of the paper is the features should also be taken into consideration.
>
> **R:** Thanks for your concerns about the core findings of our paper. We want to re-clarify the contribution of our paper. Our paper does not aim to design a new deep-learning model to achieve state-of-the-art performance in link prediction (recovery). Instead, we aim to understand when link prediction algorithms work on particular graphs and when they do not, according to important factors leading to the future link formulation. To achieve this goal, we propose a theoretical latent space model to formally understand and analyze different data factors that influence GNN's performance. By identifying important data factors and how they affect the link prediction problem together, we can provide guidance for future GNN model design and dataset selection. **Notably, taking features into consideration is not our core finding.** We add features into consideration just to provide **a more comprehensive understanding across all data factors** which can easily be applied to all graphs. Moreover, our paper majorly focuses on **GNN4LP models, the state-of-the-art models in link prediction**, rather than node embedding methods with modest performance[4].
>
>
> **Our contribution is still significant in understanding link prediction on those graphs where node features are not available.** In particular, we first exhibit the effectiveness of the local structural factor (proposition 1) and local structural factor (proposition 2). Then we find their underlying relationship: global structural proximity only shows effectiveness when local structural proximity is deficient in Lemma 1.
>
>
>
> **Weakness2 & Question:** Is the proposed latent space different from a node embedding, and if so, why? What is the difference between the latent space from the node embedding? Can you make a comparison between the latent space model and node embedding models since the latent space model and node embedding models utilize latent space and embedding, respectively, for link recovery?
>
> **Response:** Thanks for your great question about the connection between node embedding models and latent space models. Our primary answer is that they can not be directly compared as the latent space model is a **theoretical model** while the node embedding model is a **concrete graph learning algorithm**.
>
> **On the one hand,** node embedding model is a **concrete graph learning algorithm**, similar to GNN4LP models. Node embedding models aim to learn node embedding to preserve important data properties for link prediction. Nonetheless, they are not the SOTA since GNN4LP can perform much better[4]. That is the reason why we focus on GNN4LP models. **On the other hand,** the latent space model is a **theoretical model**, which helps us understand the important data factors for the link prediction task and how different link prediction algorithms work. Hence, indeed, the latent space model can be used to understand the graph embedding algorithms as well as GNNs discussed in our paper. The theoretical latent space model cannot be directly applied and learned from data to achieve satisfying performance. It serves to better understand those algorithms applied for future collection. More details on the above explanations are as follows.

---

> > ### Author Response · Authors · 2023-11-14
> > **Response to reviewer zuoh (2/2)**
> >
> > **Weakness2 & Question:(continue)** Is the proposed latent space different from a node embedding, and if so, why? What is the difference between the latent space from the node embedding? Can you make a comparison between the latent space model and node embedding models since the latent space model and node embedding models utilize latent space and embedding, respectively, for link recovery?
> >
> > **Response(continue):** The latent space model [1] is a **theoretical model** to mathematically formulate the graph data and link prediction task. Notably, our latent space model incorporates both feature and structure perspectives, providing a comprehensive mathematical description of various kinds of graph data. Such mathematical description can help us to conduct theoretical analysis and bring insights into the effectiveness of these data factors (Proposition 1,2,3) and the relationship between different factors (Lemma 1,2). **Notably, "latent space" does not correspond to latent embedding that is generally utilized in the DNN. The latent space is a typical manner of describing node pair proximity mathematically**, where nodes with a close distance in the latent space have a high probability of being connected.
> >
> > The node embedding model is a **concrete graph learning algorithm**, similar to GNN models. It transforms nodes in a graph into low-dimensional vectors, capturing the network topology and the node feature properties. This approach learns representations through data-driven algorithms, enabling effective capturing of complex patterns within the graph structure and node features for link prediction. Overall, the node embedding algorithm is to learn the important proximity information from data to recover the missing links. Notably, the capability and the performance of most node embedding models are modest compared with GNN4LP models[4]. Our paper focuses on the state-of-the-art GNN4LP models.
> >
> > The relationship between graph embedding algorithms and the latent space model is similar as to the relationship between GNNs and the latent space model. The data patterns that graph embedding algorithms try to capture can be mathematically described by the latent space model. For instance, Node2Vec[2] captures both local and global structural proximity while TADW [3] captures both local structural proximity and feature proximity. These factors are indeed captured by the latent space model. Our understanding derived from the latent space model can indeed provide insights for the both better graph embedding algorithm and GNN design to capture different important data factors simultaneously.
> >
> > The above discussion can also be found in the revision, appendix A, related work.
> >
> > [1] Peter D Hoff, Adrian E Raftery, and Mark S Handcock. Latent space approaches to social network analysis. Journal of the american Statistical association, 97(460):1090–1098, 2002.
> > [2] Grover, Aditya, and Jure Leskovec. "node2vec: Scalable feature learning for networks." Proceedings of the 22nd ACM SIGKDD international conference on Knowledge discovery and data mining. 2016.
> > [3] Network representation learning with rich text information
> > [4] Li, Juanhui, et al. "Evaluating Graph Neural Networks for Link Prediction: Current Pitfalls and New Benchmarking." NeurIPS 2023.

---

> > > ### Author Response · Authors · 2023-11-17
> > > **A kind reminder**
> > >
> > > Dear Reviewer zuoh,
> > >
> > > Thank you for taking the time to review our work. We appreciate your feedback and we have prepared a thorough response to address your concerns. We believe that we have responded to and addressed all your concerns with our revisions — in light of this, we hope you consider raising your score. Please let us know in case there are outstanding concerns, and if so, we will be happy to respond.
> > >
> > > Notably, given that we are approaching the deadline for the rebuttal phase, we hope we can have the discussion soon. Thanks！

---

> ### Author Response · Authors · 2023-11-22
> **The last day remind for reviewer zuoh**
>
> Dear reviewer zuoh, as today is the final day of our discussion, we appreciate the opportunity to engage with you. If you have any remaining questions or concerns, please don't hesitate to share them with us, we will be happy to respond. Thank you.

---

### Meta-Review · Area_Chair_BRQn · 2023-12-07

**Metareview:**

The authors explore the principles of link prediction and more specifically the role of proximity factors (three of those are identified) and their interplay. To drive the analysis, GNN for link prediction (GNN4LP) is used.

Key strengths:
- Important motivation for this work. Indeed, what is the inherent characterization of a "link" differs across datasets, and having insights to help us take it into account during modeling is valuable.
- Using latent space model analysis in this is an interesting idea with useful findings regarding relationship of proximity factors
- Experiments are convincing and collected datasets could promote further reseach

Key weaknesses:
- There has been confusion about the latent space model, some reviewers question its motivation, others question its relation with node embeddings.
- Readability and explanation of key parts needed improvement

**Justification For Why Not Higher Score:**

The scores by the reviewers overall remain borderline. While the approach is useful, the motivation and role of the latent space model still needs some further explanation in order to constitute work of maturity for a spotlight.

**Justification For Why Not Lower Score:**

I believe this paper can inspire useful future directions in the link prediction and network evolution domains. I tend to agree with the reviewer who championed the paper because of the coherent and convincing "story" of this paper, i.e. (a) focusing on GNN4LP (b) then driving the analysis (c) finding insights and (d) then suggesting instructions. Further, the use of latent space model which is also used in other network analysis domains is a good idea that can spark further research into GNNs.

---

### Decision · Program_Chairs · 2024-01-16

Accept (poster)